# High Probability Complexity Bounds for Line Search Based on Stochastic Oracles

**Billy Jin**
Cornell University
bzj3@cornell.edu

**Katya Scheinberg**
Cornell University
katyas@cornell.edu

**Miaolan Xie**
Cornell University
mx229@cornell.edu

## Abstract

We consider a line-search method for continuous optimization under a stochastic setting where the function values and gradients are available only through inexact probabilistic zeroth and first-order oracles. These oracles capture multiple standard settings including expected loss minimization and zeroth-order optimization. Moreover, our framework is very general and allows the function and gradient estimates to be biased. The proposed algorithm is simple to describe, easy to implement, and uses these oracles in a similar way as the standard deterministic line search uses exact function and gradient values. Under fairly general conditions on the oracles, we derive a high probability tail bound on the iteration complexity of the algorithm when applied to non-convex smooth functions. These results are stronger than those for other existing stochastic line search methods and apply in more general settings.

## 1 Introduction

In this paper, we analyze a line-search method when applied to the problem of minimizing an unconstrained, differentiable, possibly non-convex function $\phi : \mathbb{R}^n \to \mathbb{R}$. The goal is to find a $\varepsilon$-stationary point for $\phi$; that is, a point $x$ with $\|\nabla\phi(x)\| \leq \varepsilon$. We make the standard assumption that $\nabla\phi$ is $L$-Lipschitz, but the knowledge of $L$ is not assumed by the algorithm. We consider a setting where neither the function value $\phi(x)$ nor the gradient $\nabla\phi(x)$ are directly computable. Instead, the algorithm is given black-box access to the following probabilistic oracles:

- **Probabilistic zeroth order oracle.** Given a point $x$, the oracle computes $f(x, \xi)$, a (random) estimate of the function value $\phi(x)$. $\xi$ is a random variable (whose distribution may depend on $x$), with probability space $(\Omega, \mathcal{F}_\Omega, P)$. We assume the absolute value of the estimation error $e(x) = |f(x, \xi(x)) - \phi(x)|$ (we omit the dependence on $\xi$ for brevity) to be a "one-sided" sub-exponential-like random variable[1] with parameters $(\nu, b)$, whose mean is bounded by some constant $\epsilon_f > 0$. Specifically,

$$\mathbb{E}_\xi [e(x)] \leq \epsilon_f \text{ and } \mathbb{E}_\xi [\exp\{\lambda(e(x) - \mathbb{E}[e(x)])\}] \leq \exp\left(\frac{\lambda^2 \nu^2}{2}\right), \quad \forall \lambda \in \left[0, \frac{1}{b}\right]. \quad (1)$$

- **Probabilistic first order oracle.** Given a point $x$ and a constant $\alpha > 0$, the oracle computes $g(x, \xi')$, a (random) estimate of the gradient $\nabla\phi(x)$, such that

$$\mathbb{P}_{\xi'} \left(\|g(x, \xi') - \nabla\phi(x)\| \leq \max\{\epsilon_g, \kappa\alpha\|g(x, \xi')\|\}\right) \geq 1 - \delta. \quad (2)$$

Here, $\xi'$ is a random variable (whose distribution may depend on $x$), with associated probability space $(\Omega', \mathcal{F}_{\Omega'}, P')$. $(1 - \delta) \in (0, 1)$ is the probability, intrinsic to the oracle, that the

---

[1]This is a weaker requirement than assuming $e(x)$ to be sub-exponential, as one only needs to guard against the possibility of the error $e(x)$ being too large.

35th Conference on Neural Information Processing Systems (NeurIPS 2021).

gradient estimate is "sufficiently accurate" with respect to $\epsilon_g, \kappa$, and $\alpha$. Lastly, $\kappa, \epsilon_g \geq 0$ are constants, intrinsic to the oracle, which represent the precision the oracle can achieve. Note that $\epsilon_g$ allows the gradient estimate to be bounded away from the true gradient by a constant distance.

**Remark**    We will analyze a line search algorithm that relies on these two oracles. In the zeroth order oracle, the constants $\epsilon_f$ and $(\nu, b)$ are intrinsic. In the first order oracle, $\kappa, \epsilon_g$, and $\delta$ are intrinsic. These values cannot be controlled. On the other hand, $\alpha$ is an *input* to the first order oracle that can be chosen by the algorithm. In fact, as we shall see in Section 3, $\alpha$ will be the step size of the line search method.

These two oracles cover several settings, including

- Standard supervised learning, where gradients and values of the loss function are computed based on a mini-batch. Here, the random variables $\xi$ and $\xi'$ in the zeroth and first order oracles represent the random set of samples in the mini-batch.
- Zeroth order optimization, where gradients are estimated via randomized finite differences using (possibly noisy) function values. This arises in policy gradients in reinforcement learning, as is used in [SHC+16] and analyzed in [BCCS21].
- A variety of other settings, where the gradients and function estimates may be biased stochastic estimates of the true gradients and function values.

The constants in the oracles determine the precision of the function and gradient estimates. These constants will also dictate the accuracy achievable by the line search method we analyze. Specifically, if $\epsilon_f = 0$ and $\epsilon_g = 0$, then the algorithm converges to a stationary point. Otherwise, a precise lower bound is derived for the smallest $\|\nabla\phi(x)\|$ the algorithm can achieve, in terms of the constants in the oracles. It is worth noting that the oracles can be biased. Indeed, the zeroth order oracle can incur arbitrarily large error, as long as it satisfies 1. Moreover, the first order oracle only requires $g(x, \xi'(x))$ to be a "sufficiently accurate" estimate of $\nabla\phi(x)$ with probability $1-\delta$. Thus $g(x, \xi'(x))$ can be an arbitrary vector with probability $\delta$, so it in principle can have an arbitrarily large bias.

The line-search algorithm is given in Section 3. It is a modification of the standard Armijo-based line search algorithm [NW06], with access to the zeroth and first order oracles. The two small modifications are: 1) The Armijo condition is relaxed by an additive constant $2\epsilon_f$, to account for the inexact function evaluations, and 2) The first order oracle is called in each iteration, and a new search direction is generated whenever the step size changes. This allows the method to progress to near-stationary points without assuming the gradient estimates (e.g. the mini-batch gradients in supervised learning) to be Lipschitz continuous.

Our framework and analysis are based on results in [CS17], [GRVZ18] and [BCS19]. However, there are several key differences. In [CS17] and [BCS19] the line search has access to stronger oracles, with $\epsilon_g = 0$ and $|f(x,\xi) - \phi(x)| \leq \epsilon_f$ deterministically. Under these assumptions, [CS17] and [BCS19] derive an *expected* iteration complexity bound. In this paper, we provide a *high probability* tail bound on the iteration complexity, showing that the algorithm is very likely to succeed in a number of iterations on the order of its expected iteration complexity. Moreover, we consider more general oracles, with arbitrary $\epsilon_g$ and possibly unbounded $|f(x,\xi) - \phi(x)|$. Thus, we significantly strengthen the results in [CS17] and [BCS19]. To the best of our knowledge, the only other high probability complexity bound of this kind is derived in [GRVZ18] for a trust region algorithm under the assumption $\epsilon_g = 0$ and $|f(x,\xi) - \phi(x)| = 0$ deterministically, which are much stronger oracles.

Stochastic line search has also been analyzed in [PS20] and [VML+19]. In [PS20] the assumptions on $|f(x,\xi) - \phi(x)|$ are different. On the one hand, they allow for more general distributions than sub-exponential. On the other hand, it is assumed that $|f(x,\xi) - \phi(x)|$ can be made arbitrarily small with some fixed probability. An expected iteration complexity bound is then derived for arbitrarily small $\varepsilon$. In contrast, we do not assume this, and analyze the iteration complexity of reaching an $\varepsilon$-stationary point, with $\varepsilon$ lower-bounded by a function of the constants in the oracles. Moreover, our analysis and results are much simpler than those in [PS20] and we derive an iteration complexity bound in high probability, not just in expectation.

In [VML+19], the traditional line search is analyzed for empirical loss minimization, where the function oracles are implemented using a random mini-batch of a fixed size. The mini-batch remains

fixed during backtracking until a standard Armijo condition is satisfied. Thus the search direction remains the same until a step is taken. While good computational performance has been reported in [VML+19], its theoretical analysis requires several very restrictive assumptions, especially for non-convex functions. Also, they bound the expected sum of squared gradient norms, while we bound the iteration complexity with high probability. We note that using similar techniques as in [BCS19], our analysis can be extended to the convex and strongly convex cases.

**In summary**, we present an analysis of an adaptive line search algorithm under very general conditions on the gradient and function estimates. The results not only subsume most results in the prior literature, but also substantially extend the framework. Moreover, high probability tail bounds on iteration complexity are derived, instead of only expected iteration complexity.

## 2 Oracles

In this section, we discuss a couple of settings, and show how they are captured by our framework. All norms used are 2-norm.

### 2.1 Expected loss minimization

Let us first discuss how the oracle definitions apply to expected loss minimization. In this setting, $\phi(x) = \mathbb{E}_{d\sim\mathcal{D}}[\ell(x,d)]$. Here, $x$ is the model parameters, $d$ is a data sample following distribution $\mathcal{D}$, and $\ell(x,d)$ is the loss when the model parameterized by $x$ is evaluated on data point $d$.

In this case, the zeroth and first order oracles can be as follows, where $\mathcal{S}$ is a mini-batch sampled from $\mathcal{D}$:

$$f(x,\mathcal{S}) = \frac{1}{|\mathcal{S}|}\sum_{d\in\mathcal{S}}\ell(x,d), \quad g(x,\mathcal{S}) = \frac{1}{|\mathcal{S}|}\sum_{d\in\mathcal{S}}\nabla_x\ell(x,d). \tag{3}$$

In general, $\mathcal{S}$ can be chosen to depend on $x$. We now show how our zeroth and first order oracle conditions are satisfied by selecting an appropriate sample size $|\mathcal{S}|$.

**Proposition 1.** *Let $\hat{e}(x,d) := \ell(x,d) - \phi(x)$ be a $(\hat{\nu}(x), \hat{b}(x))$-subexponential random variable and $Var_{d\sim\mathcal{D}}[\ell(x,d)] \leq \hat{\epsilon}(x)^2$, for some $\hat{\nu}(x), \hat{b}(x), \hat{\epsilon}(x)$. Let $e(x,\mathcal{S}) = |f(x,\mathcal{S}) - \phi(x)|$ and $N = |\mathcal{S}|$, then*

$$\mathbb{E}_{\mathcal{S}}[e(x,\mathcal{S})] \leq \frac{1}{\sqrt{N}}\hat{\epsilon}(x) \quad and \quad e(x,\mathcal{S}) \text{ is } (\nu(x), b(x))\text{-subexponential,}$$

*with $\nu(x) = b(x) = 8e^2\max\left\{\frac{\hat{\nu}(x)}{\sqrt{N}}, \hat{b}(x)\right\}$.*

In the case when the support of $\mathcal{D}$ is bounded, $\ell$ is Lipschitz, and the set of $x$ we consider is bounded, the assumption of Proposition 1 is satisfied. Thus, $f(x,\mathcal{S})$ is a zeroth order oracle with $\epsilon_f = \sup_x \frac{1}{\sqrt{N}}\hat{\epsilon}(x)$, $\nu = \sup_x \nu(x)$, and $b = \sup_x b(x)$, and $\epsilon_f$ can be made arbitrarily small by taking a large enough sample.

Under standard assumptions on $\nabla\ell(x,d)$, for instance, suppose Assumption 4.3 in [BCN18] holds: for some $M_c, M_v \geq 0$ and for all $x$,

$$\mathbb{E}_{d\sim\mathcal{D}}\|\nabla\ell(x,d) - \nabla\phi(x)\|^2 \leq M_c + M_v\|\nabla\phi(x)\|^2, \tag{4}$$

one can show $g(x,\mathcal{S})$ is a first order oracle with a large enough sample size.

**Proposition 2.** *Let $g = g(x,\mathcal{S})$. Assuming $\mathbb{E}_{d\sim\mathcal{D}}\nabla\ell(x,d) = \nabla\phi(x)$, then*

$$|\mathcal{S}| \geq \frac{M_c + M_v\|\nabla\phi(x)\|^2}{\delta}\min\left\{\frac{1}{\epsilon_g^2}, \frac{(1+\kappa\alpha)^2}{\kappa^2\alpha^2\|\nabla\phi(x)\|^2}\right\}$$

*implies*

$$\mathbb{P}\left(\|g - \nabla\phi(x)\| \leq \max\{\epsilon_g, \kappa\alpha\|g\|\}\right) \geq 1 - \delta.$$

*This bound implies a looser bound of:*

$$|\mathcal{S}| \geq \max\left\{\frac{2M_c}{\delta\epsilon_g^2}, \frac{2M_v(1+\kappa\alpha)^2}{\delta\kappa^2\alpha^2}\right\}.$$

**Remark** Let us discuss what is required from the minibatch size in this setting. Unlike standard SGD, the minibatch size is chosen dynamically. When convergence to a stationary point is desired, $\epsilon_g$ has to be zero, and gradient estimate $g_k$ tends to zero. Thus, $\|g - \nabla\phi(x)\|$ also has to tend to zero. If $M_c = 0$, then fixing the mini-batch size to be at least $\frac{M_v(1+\kappa\alpha)^2}{\delta\kappa^2\alpha^2}$ provides a valid first order oracle. Thus, unless $\alpha$ tends to zero, the minibatch size remains bounded from below. This is similar to the interpolation condition used in e.g. [BM11, VML$^+$19]. On the other hand, when $M_c > 0$, the minibatch size has to grow in order to approach a stationary point. This is similar to dynamic minibatch size selection, discussed, e.g. in [BCNW12]. The difference between our results and those in [BCNW12] is that our batch bound is implementable and guarantees convergence, while the one in [BCNW12] is implementable only as a heuristic. As our computational results show, however, a fixed and small mini-batch size appears to work very well, perhaps because $M_c$ is small.

## 2.2 Randomized finite difference gradient approximation

Gradient estimates based on randomized finite differences using noisy function evaluations have become popular for zeroth order optimization, particularly for model-free policy optimization in reinforcement learning [SHC$^+$16, FGKM18].

In this setting, the zeroth order oracle is assumed to be available, but with a more strict assumption that $e(x) \le \epsilon_f$ deterministically. The first order oracle is obtained using the zeroth order oracle as follows. Let $\mathcal{U} = \{u_i : i = 1, \ldots, |\mathcal{U}|\}$ be a set of random vectors, with each vector following some "nice" distribution (e.g. standard Gaussian). Then,

$$g(x, \mathcal{U}) = \sum_{i=1}^{|\mathcal{U}|} \frac{f(x + \sigma u_i, \xi) - f(x, \xi)}{\sigma |\mathcal{U}|} u_i, \tag{5}$$

where $\sigma$ is the sampling radius. The proposition below shows that (5) with a large enough sample size gives a first order oracle.

**Proposition 3.** *Let $g = g(x, \mathcal{U})$, and fix $\epsilon_g = 2\left(\sqrt{n}L\sigma + \frac{\sqrt{n}\epsilon_f}{\sigma}\right)$ where $n$ is the dimension of $x$. Then*

$$|\mathcal{U}| \ge \frac{\frac{3}{4}L^2\sigma^2 n(n+2)(n+4) + \frac{12\epsilon_f^2}{\sigma^2}n + 18n\|\nabla\phi(x)\|^2}{\delta} \min\left\{\frac{4}{\epsilon_g^2}, \frac{1}{\left(\frac{\kappa\alpha}{1+\kappa\alpha}\|\nabla\phi(x)\| - \frac{\epsilon_g}{2}\right)^2}\right\}$$

*implies*

$$\mathbb{P}\left(\|g - \nabla\phi(x)\| \le \max\{\epsilon_g, \kappa\alpha\|g\|\}\right) \ge 1 - \delta.$$

*Note that in the setting, $\epsilon_g$ is a fixed bias dependent on $\sigma$, and cannot be made arbitrarily small.*

**Remark** Note that $\epsilon_g$ defines the neighborhood of convergence for any method that relies on this oracle, and the smallest value for $\epsilon_g$ is achieved by setting $\sigma = \mathcal{O}(\sqrt{\epsilon_f})$. Let us now discuss the minibatch size. Under the assumption that $\epsilon_f$ is small, $\frac{3}{4}L^2\sigma^2 n(n+2)(n+4) + \frac{12\epsilon_f^2}{\sigma^2}$ is also small. Thus when $\|\nabla\phi(x)\|$ is larger than or on the order of $\epsilon_g$, then the sample set size remains constant and is proportional to $n$. In [NS17] a constant step size stochastic gradient descent is applied using sample size $|\mathcal{U}| = 1$, thus each step requires about $n$ fewer samples. However, the step size has to be roughly $n$ times smaller to account for the variance of the stochastic oracles based on one sample, thus the overall complexity is the same.

Other finite difference approximation schemes and their centralized versions (see [BCCS21] for a reference on these) also give suitable first order oracles.

## 2.3 Other settings

Our oracle framework also fits a variety of other settings, as we allow the randomness $\xi$ and $\xi'$ of the zeroth and first order oracles to be dependent on $x$ and on each other, possibly following different distributions. Moreover, the oracles allow the function and gradient estimations to be arbitrarily bad occasionally, which allows them to capture settings where measurements are corrupted with outliers. The exact derivations of these oracles in these different settings are subjects of future exploration.

# 3 Algorithm and notation

We consider the line search algorithm proposed by [BCS19], which is an extension of the line search algorithm in [CS17] to the setting of inexact function estimates. In both algorithms, a random gradient estimate is used to attempt a step. We name the algorithm "ALOE", which stands for Adaptive Line-search with Oracle Estimations. Compared to [CS17], the key modification of the algorithm is the relaxation of the Armijo condition by an additive constant $2\epsilon_f$. The difference between this algorithm and the more standard line search methods such as the ones in [NW06] and [VML+19] is that the gradient estimate is recomputed in each iteration, whether or not a step is accepted. Note that all input parameters are user controlled, except for $\epsilon_f$. In fact, the input $\epsilon_f$ here is only required to be some upper bound for $\mathbb{E}[e(x)]$, not necessarily the tightest one. Moreover, our computational results in Section 6 indicate that estimating $\epsilon_f$ is relatively easy in practice, and the algorithm is robust to the choice of $\epsilon_f$.

---

**Algorithm 1** Adaptive Line-search with Oracle Estimations (ALOE)

---

**Input:** Parameter $\epsilon_f$ of the zeroth order oracle, starting point $x_0$, max step size $\alpha_{\max} > 0$, initial step size $\alpha_0 < \alpha_{\max}$, constants $\theta, \gamma \in (0, 1)$.

1: **for** $k = 0, 1, 2, \ldots$ **do**
2:     **Compute gradient approximation $g_k$:**
        Generate the direction $g_k = g(x_k, \xi_k')$ using the probabilistic first order oracle, with $\alpha = \alpha_k$.
3:     **Check sufficient decrease:**
        Let $x_k^+ = x_k - \alpha_k g_k$. Generate $f(x_k, \xi_k)$ and $f(x_k^+, \xi_k^+)$ using the probabilistic zeroth order oracle. Check the modified *Armijo* condition:

$$f(x_k^+, \xi_k^+) \leq f(x_k, \xi_k) - \alpha_k \theta \|g_k\|^2 + 2\epsilon_f. \tag{6}$$

4:     **Successful step:**
        If (6) holds, then set $x_{k+1} \leftarrow x_k^+$ and $\alpha_{k+1} \leftarrow \min\{\alpha_{\max}, \gamma^{-1}\alpha_k\}$.
5:     **Unsuccessful step:**
        Otherwise, set $x_{k+1} \leftarrow x_k$ and $\alpha_{k+1} \leftarrow \gamma\alpha_k$.

---

In this paper we impose the following standard assumption on $\phi(x)$.

**Assumption 1.** $\nabla\phi$ is *L-Lipschitz smooth and $\phi$ is bounded from below by some constant $\phi^*$.*

Let $e_k = |f(x_k, \xi_k) - \phi(x_k)|$ and $e_k^+ = |f(x_k^+, \xi_k^+) - \phi(x_k^+)|$. Recall that $e_k$ and $e_k^+$ satisfy (1) from the definition of the zeroth order oracle. We will consider two cases; 1) $e_k$ and $e_k^+$ are deterministically bounded by $\epsilon_f$, in which case $\nu$ and $b$ in (1) can be chosen to be 0, and 2) $\nu$ and $b$ are not necessarily zero, in which case we assume the random variables $e_k + e_k^+$ are all independent.

**Assumption 2.** *Either $e_0, e_0^+, e_1, e_1^+, \ldots$ are all deterministically bounded by $\epsilon_f$, or the random variables $\{e_0 + e_0^+, e_1 + e_1^+, \ldots\}$ are independent.*

**Definition 1** (Definition of a true iteration). *We say an iteration $k$ is **true** if*

$$\|g_k - \nabla\phi(x_k)\| \leq \max\{\epsilon_g, \kappa\alpha_k\|g_k\|\} \quad and \quad e_k + e_k^+ \leq 2\epsilon_f,$$

*and **false** otherwise.*

Let $M_k$ denotes the triple $\{\Xi_k, \Xi_k^+, \Xi_k'\}$, whose realizations are $\{\xi_k, \xi_k^+, \xi_k'\}$. Algorithm 1 generates a stochastic process adapted to the filtration $\{\mathcal{F}_k : k \geq 0\}$, where $\mathcal{F}_k = \sigma(M_0, M_1, \ldots, M_k)$. We define the following random variables, measurable with respect to $\mathcal{F}_k$.

- $I_k := \mathbb{1}\{\text{iteration } k \text{ is true}\}$.
- $\Theta_k := \mathbb{1}\{\text{iteration } k \text{ is successful}\}$.
- $T_\varepsilon := \min\{k : \|\nabla\phi(x_k)\| \leq \varepsilon\}$, the iteration complexity of the algorithm for reaching $\varepsilon$-stationarity.
- $Z_k := \phi(x_k) - \phi^* \geq 0$, a measure of progress.

It is easy to see that $T_\varepsilon$ is a *stopping time* of the stochastic process with respect to $\mathcal{F}_k$. We derive a high probability tail bound for $T_\varepsilon$, and obtain an iteration complexity bound in high probability for Algorithm 1 when applied to non-convex functions. The final result is summarized below with simplified constants. The full statement is in Theorem 4.

**Theorem 1** (Main convergence result with simplified constants)**.** *Suppose Assumptions 1 and 2 hold, and (for simplicity) $\theta = \frac{1}{2}$, $\alpha_{\max} \geq 1$ and $\kappa \geq \max\{L, 1\}$. Then, for any*

$$\varepsilon \geq 4 \max \left\{ \epsilon_g, (1 + \kappa \alpha_{\max}) \sqrt{(L + 2\kappa)\epsilon_f} \right\},$$

*we have the following bound on iteration complexity:*

*For any $s \geq 0$, $p = 1 - \delta - e^{-\min\{\frac{u^2}{2\nu^2}, \frac{u}{2b}\}}$, $\hat{p} \in (\frac{1}{2} + \frac{4\epsilon_f + s}{C\varepsilon^2}, p)$, and $t \geq \frac{R}{\hat{p} - \frac{1}{2} - \frac{4\epsilon_f + s}{C\varepsilon^2}}$,*

$$\mathbb{P}\left(T_\varepsilon \leq t\right) \geq 1 - \exp\left(-\frac{(p - \hat{p})^2}{2p^2}t\right) - \exp\left(-\min\left\{\frac{s^2 t}{8\nu^2}, \frac{st}{4b}\right\}\right).$$

*Here, $u = \inf_x\{\epsilon_f - \mathbb{E}[e(x)]\}$, $R = \frac{\phi(x_0) - \phi^*}{C\varepsilon^2} - \frac{\ln((L+2\kappa)\alpha_0)}{\ln\gamma}$, and $C = \frac{1}{2(L+2\kappa)(1+\kappa\alpha_{\max})^2}$.*

**Remark** This theorem essentially shows that the iteration complexity of Algorithm 1 is bounded by a quantity on the order of

$$\frac{1}{p - \frac{1}{2} - \frac{4\epsilon_f + s}{C\varepsilon^2}} \left( \frac{\phi(x_0) - \phi^*}{C\varepsilon^2} - \frac{\ln((L+2\kappa)\alpha_0)}{\ln\gamma} \right)$$

with overwhelmingly high probability. If $p = 1$ and $\epsilon_f = 0$, the above quantity essentially recovers the iteration complexity of the deterministic line search algorithm.

# 4  Analysis framework for the high probability bound

In this section we present the main ideas underlying the theoretical analysis. We first state general conditions on the stochastic process (Assumption 3), from which we are able to derive a high probability tail bound on the iteration complexity. They are listed as assumptions here, and in the next section, we will show that they indeed hold for Algorithm 1 when applied to non-convex smooth functions $\phi$.

**Assumption 3** (Properties of the stochastic process)**.** *There exist a constant $\bar{\alpha} > 0$ and a non-decreasing function $h : \mathbb{R} \to \mathbb{R}$, which satisfies $h(\alpha) > 0$ for any $\alpha > 0$, such that for any realization of the algorithm, the following hold for all $k < T_\varepsilon$:*

- *(i) $h(\bar{\alpha}) > 8\epsilon_f$.*

- *(ii) $\mathbb{P}(I_k = 1 \mid \mathcal{F}_{k-1}) \geq p$ for all $k$, with some $p \in (\frac{1}{2} + \frac{4\epsilon_f}{h(\bar{\alpha})}, 1]$.*

- *(iii) If $I_k\Theta_k = 1$ then $Z_{k+1} \leq Z_k - h(\alpha_k) + 4\epsilon_f$. (True, successful iterations make progress.)*

- *(iv) If $\alpha_k \leq \bar{\alpha}$ and $I_k = 1$ then $\Theta_k = 1$.*

- *(v) $Z_{k+1} \leq Z_k + 2\epsilon_f + e_k + e_k^+$ for all $k$.*

The following key lemma follows easily from Assumption 3 (ii) and the Azuma-Hoeffding inequality [Azu67] applied to the submartingale $\sum_{k=0}^{t-1} I_k - pt$.

**Lemma 1.** *For all $1 \leq t \leq T_\varepsilon$, and any $\hat{p} \in [0, p)$, we have*

$$\mathbb{P}\left(\sum_{k=0}^{t-1} I_k \leq \hat{p}t\right) \leq \exp\left(-\frac{(p - \hat{p})^2}{2p^2}t\right).$$

We now define another indicator variable that will be used in the analysis.

**Definition 2** (Large step). *For all integers $k \geq 0$, define the random variable $U_k$ as follows:*

$$U_k = \begin{cases} 1, & \text{if } \min\{\alpha_k, \alpha_{k+1}\} \geq \bar{\alpha}, \\ 0, & \text{if } \max\{\alpha_k, \alpha_{k+1}\} \leq \bar{\alpha}. \end{cases}$$

*We will say that step $k$ is a **large step** if $U_k = 1$. Otherwise, step $k$ is a **small step**.*

By the dynamics of the process, every step is either a large step or a small step, but not both.

Our analysis will rely on the following key observation: By Assumption 3, if iteration $k$ has $U_k \Theta_k I_k = 1$, then $Z_k$ gets reduced by at least $h(\bar{\alpha}) - 4\epsilon_f > 0$. We call such an iteration a "*good*" iteration, because it makes progress towards optimality by at least a fixed amount. On the other hand, on any other iteration $k$, $Z_k$ can increase by at most $2\epsilon_f + e_k + e_k^+$. The idea of the analysis is to show that with high probability, the progress made by the good iterations dominates the damage caused by the other iterations. The crux of the proof is to show that with high probability, a large enough constant fraction of the iterations are good (up to another additive constant).

The following key lemma is the engine of the analysis. It shows that if the stopping time has not been reached and a large enough number of iterations are true, then there must be a large number of good iterations.

**Lemma 2.** *For any positive integer $t$ and any $\hat{p} \in (\frac{1}{2}, 1]$, we have*

$$\mathbb{P}\left( T_\varepsilon > t \text{ and } \sum_{k=0}^{t-1} I_k \geq \hat{p}t \text{ and } \sum_{k=0}^{t-1} U_k \Theta_k I_k < \left(\hat{p} - \frac{1}{2}\right)t - \frac{d}{2} \right) = 0,$$

*where $d = \max\left\{-\frac{\ln \alpha_0 - \ln \bar{\alpha}}{\ln \gamma}, 0\right\}$.*

### 4.1 Bounded noise case

In [CS17] and [BCS19], the *expected* iteration complexity of the line search algorithm is bounded under the assumptions that $e(x) = 0$ and $e(x) \leq \epsilon_f$ for all $x$, respectively. We now derive a *high probability* tail bound on the iteration complexity under the assumption that $e(x) \leq \epsilon_f$ for all $x$. Note that we do not need to assume that the errors $e(x)$ are independent in the bounded noise setting. Thus, this analysis applies even when the noise is deterministic or adversarial.

Under Assumption 3 in the bounded noise setting, we have $Z_{k+1} \leq Z_k + 4\epsilon_f$ in all iterations, and $Z_{k+1} \leq Z_k - h(\bar{\alpha}) + 4\epsilon_f$ in good iterations. Putting this together with Lemma 2 and the other conditions in Assumption 3, we obtain the following theorem.

**Theorem 2** (Iteration complexity in the bounded noise setting). *Suppose Assumption 3 holds, and $e_k, e_k^+ \leq \epsilon_f$ at every iteration. Then for any $\hat{p} \in (\frac{1}{2} + \frac{4\epsilon_f}{h(\bar{\alpha})}, p)$, and $t \geq \frac{R}{\hat{p} - \frac{1}{2} - \frac{4\epsilon_f}{h(\bar{\alpha})}}$ we have*

$$\mathbb{P}\left(T_\varepsilon \leq t\right) \geq 1 - \exp\left(-\frac{(p - \hat{p})^2}{2p^2}t\right),$$

*where $R = \frac{Z_0}{h(\bar{\alpha})} + \frac{d}{2}$ and $d = \max\left\{-\frac{\ln \alpha_0 - \ln \bar{\alpha}}{\ln \gamma}, 0\right\}$.*

### 4.2 General sub-exponential noise case

We now present a high probability bound for the iteration complexity with general sub-exponential noise in the zeroth order oracle. The result is very similar to that of Theorem 2. The main difference from the bounded noise analysis is that instead of bounding the "damage" caused on a per-iteration basis, we bound the sum of all such damages over all iterations. The fact that the noises are sub-exponential and independent allows us to apply Bernstein's inequality to obtain an upper bound on this sum that holds with high probability.

**Theorem 3** (Iteration complexity in the sub-exponential noise setting). *Suppose Assumptions 2 and 3 hold. Then for any $s \geq 0$, $\hat{p} \in (\frac{1}{2} + \frac{4\epsilon_f + s}{h(\bar{\alpha})}, p)$, and $t \geq \frac{R}{\hat{p} - \frac{1}{2} - \frac{4\epsilon_f + s}{h(\bar{\alpha})}}$, we have*

$$\mathbb{P}\left(T_\varepsilon \leq t\right) \geq 1 - \exp\left(-\frac{(p - \hat{p})^2}{2p^2}t\right) - e^{-\min\left\{\frac{s^2 t}{8\nu^2}, \frac{st}{4b}\right\}},$$

*where $R = \frac{Z_0}{h(\bar{\alpha})} + \frac{d}{2}$ and $d = \max\left\{-\frac{\ln \alpha_0 - \ln \bar{\alpha}}{\ln \gamma}, 0\right\}$.*

# 5 Final iteration complexity of the line search algorithm

In the previous section, we presented high probability tail bounds on the iteration complexity, assuming Assumption 3 holds. We now verify that Assumption 3 indeed holds for Algorithm 1 when applied to smooth functions. Together with the results in Section 4, this allows us to derive an explicit high-probability bound on the iteration complexity.

As noted earlier, when either $\epsilon_f$ or $\epsilon_g$ are not zero, Algorithm 1 does not converge to a stationary point, but converges to a neighborhood where $\|\nabla\phi(x)\| \leq \varepsilon$, with $\varepsilon$ bounded from below in terms of $\epsilon_f$ or $\epsilon_g$. The specific relationship is as follows.

**Inequality 1** (Lower bound on $\varepsilon$).

$$\varepsilon > \max\left\{\frac{\epsilon_g}{\eta}, \max\left\{1 + \kappa\alpha_{\max}, \frac{1}{1-\eta}\right\} \cdot \sqrt{\frac{4\epsilon_f}{\theta(p-\frac{1}{2})} \cdot \max\left\{\frac{0.5L+\kappa}{1-\theta}, \frac{L(1-\eta)}{2(1-2\eta-\theta(1-\eta))}\right\}}\right\},$$

*for some $\eta \in (0, \frac{1-\theta}{2-\theta})$.*

Here $\eta$ can be any value in the interval. $p = 1 - \delta$ when in the bounded noise setting, and $p = 1 - \delta - \exp\left(-\min\{\frac{u^2}{\nu^2}, \frac{u}{b}\}\right)$ otherwise, with $u = \inf_x\{\epsilon_f - \mathbb{E}[e(x)]\}$.

**Proposition 4** (Assumption 3 holds for Algorithm 1). *If Inequality 1 and Assumption 1 and 2 hold, then Assumption 3 holds for Algorithm 1 with the following $p$, $\bar{\alpha}$ and $h(\alpha)$:*

1. $p = 1 - \delta$ *when the noise is bounded by $\epsilon_f$, and $p = 1 - \delta - \exp\left(-\min\{\frac{u^2}{2\nu^2}, \frac{u}{2b}\}\right)$ otherwise. Here $u = \inf_x\{\epsilon_f - \mathbb{E}[e(x)]\}$.*

2. $\bar{\alpha} = \min\left\{\frac{1-\theta}{0.5L+\kappa}, \frac{2(1-2\eta-\theta(1-\eta))}{L(1-\eta)}\right\}$.

3. $h(\alpha) = \min\left\{\frac{\theta\epsilon^2\alpha}{(1+\kappa\alpha_{\max})^2}, \theta\alpha(1-\eta)^2\epsilon^2\right\}$.

Applying Theorem 3 now gives the explicit complexity bound for Algorithm 1.

**Theorem 4.** *Suppose the Inequality 1 on $\varepsilon$ is satisfied for some $\eta \in (0, \frac{1-\theta}{2-\theta})$, and Assumptions 1 and 2 hold, then we have the following bound on the iteration complexity: For any $s \geq 0$, $\hat{p} \in (\frac{1}{2} + \frac{4\epsilon_f+s}{C\varepsilon^2}, p)$, and $t \geq \frac{R}{\hat{p}-\frac{1}{2}-\frac{4\epsilon_f+s}{C\varepsilon^2}}$,*

$$\mathbb{P}\left(T_\varepsilon \leq t\right) \geq 1 - \exp\left(-\frac{(p-\hat{p})^2}{2p^2}t\right) - \exp\left(-\min\left\{\frac{s^2t}{8\nu^2}, \frac{st}{4b}\right\}\right).$$

*Here, $R = \frac{\phi(x_0)-\phi^*}{C\varepsilon^2} + \max\left\{-\frac{\ln\alpha_0-\ln\bar{\alpha}}{\ln\gamma}, 0\right\}$, $C = \min\left\{\frac{1}{(1+\kappa\alpha_{\max})^2}, (1-\eta)^2\right\}\bar{\alpha}\theta$, with $p$ and $\bar{\alpha}$ as defined in Proposition 4.*

**Remark** Inequality 1 makes sure there exists some $\hat{p} \in (\frac{1}{2} + \frac{4\epsilon_f+s}{C\varepsilon^2}, p)$ for some $s > 0$. The above theorem is for the general sub-exponential noise setting. In the bounded noise special case, we have $s = 0$, and the last term $\exp\left(-\min\left\{\frac{s^2t}{8\nu^2}, \frac{st}{4b}\right\}\right)$ in the probability is not present.

# 6 Experiments

In this section, we illustrate that the proposed stochastic algorithm ALOE can be at least as efficient in practice as the line search in [VML$^+$19], and much more efficient than full gradient line search. From the experiments, we show that estimating $\epsilon_f$ is not difficult, and taking mini-batches of a fixed size indeed provides good zeroth and first order oracles in practice.

For illustration, we first conduct experiments on all the datasets for binary classification with 150 to 5000 data points from the Penn Machine Learning Benchmarks repository (PMLB) [RLLC$^+$21]. In total, there are 64 such datasets. Each binary classification problem is formulated as a logistic

regression problem with an RBF kernel (with parameter $\sigma = 1$). All experiments were conducted on a 2020 MacBook Pro with an M1 chip and 16GB of memory.

We compare the following three algorithms, and they are implemented as follows.

- **ALOE.** The zeroth and first order oracles are implemented using the same mini-batch of a fixed size within each iteration. Batch sizes are taken to be 128. We estimate $\epsilon_f$ at the beginning of every epoch (i.e. every $K$ iterations, where $K$ equals the total number of data samples divided by 128), by computing $\frac{1}{5}$ times the empirical standard deviation of 30 zeroth order oracle calls with batch size 128 at the current point. We found in practice the algorithm is quite robust to how $\epsilon_f$ is chosen. The relevant plots are in Appendix F. The parameter we used are $\gamma = 0.8$, $\theta = 0.2$, $\alpha_0 = 1$ and $\alpha_{\max} = 10$.

- **SLS.** The SLS algorithm (also called "SGD + Armijo") proposed in [VML$^+$19] differs from ALOE in that $\epsilon_f = 0$ and that the same mini-batch is used while backtracking until the Armijo condition is satisfied. We implemented the algorithm using mini-batch size 128 and the parameters suggested in their paper. We tried various parameter combinations for SLS and found the performance of the suggested parameters to work best.

- **Full gradient line search.** The full gradient line search algorithm is implemented using the entire dataset for function and gradient evaluations on each iteration. Taking $\epsilon_f = 0$ and the other parameters are the same as used in ALOE. For fair comparison in our experiments, we allow full gradient line search to make the same number of passes over each dataset as ALOE.

We conducted 5 trials for each dataset and ran each algorithm with initial points taken randomly from a standard Gaussian distribution. In Figure 1 we compare the overall performance of the three algorithms in the following way. For each dataset and algorithm, the average best value is defined as the average of the minimum training loss attained over 5 different trials. For each dataset we record the difference between the average best values achieved by SLS vs. ALOE, and plot the resulting 64 numbers as a histogram. The same is done for full gradient line search vs. ALOE. See Figure 1. Under this metric, ALOE achieves better training loss than SLS algorithm in 62 out of 64 datasets, and is always better than the full gradient line search.

Figure 2 illustrates the decay of training losses using these three algorithms for three datasets. In many cases ALOE decreases the training loss more rapidly than the other two algorithms. Testing set accuracy comparisons are also carried out, using random $80 : 20$ splits of datasets, as shown in Figure 3. Test accuracy is defined as the proportion of data points in the testing set classified correctly. The results show that ALOE is competitive in terms of test accuracy as well. More performance and test accuracy plots for different datasets, models and loss functions are in Appendix.

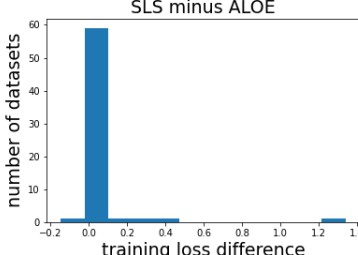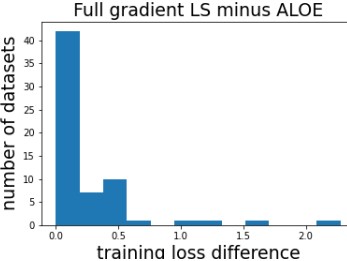

Figure 1: The performance of ALOE is consistently better (above 0) compared to SLS and full gradient line search over the 64 datasets, with a bigger advantage over full gradient line search.

## 7 Final Remarks

We conclude the paper with a brief overview of our theoretical results with respect to those in [VML$^+$19]. The stochastic line search in [VML$^+$19] is proposed specifically for empirical risk minimization, and the zeroth and first order are implemented using mini-batch of a fixed size. The same mini-batch is used for all consecutive unsuccessful iterations. This guarantees that a successful iteration is eventually achieved for Armijo condition with $\epsilon_f = 0$, under the assumption

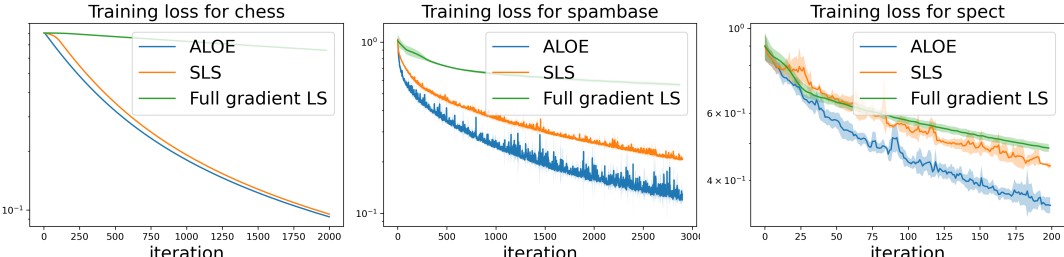

Figure 2: The training loss decays of three algorithms.

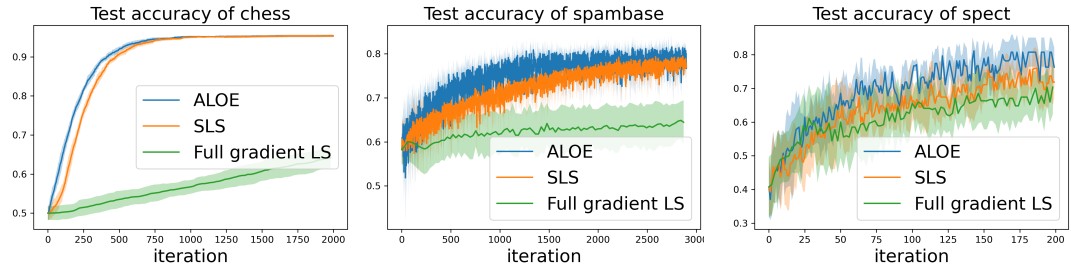

Figure 3: The testing accuracy improvements of three algorithms.

that for every mini-batch, $g(x, \xi')$ is Lipschitz continuous. The convergence analysis then assumes that $M_c = 0$ in (4) (*strong growth condition*) and in the case when $\phi$ is not convex, the step size parameter is bounded above by $\frac{1}{LM_v}$. Thus, the method itself and its convergence are not better than those of a stochastic gradient descent with a fixed step size bounded by $\frac{1}{LM_v}$ [BCN18]. It is also assumed that the step size is reset to a fixed value at the start of each iteration, which is impractical. Good computational results are reported in [VML$^+$19] for a heuristic version of the algorithm where the restrictions of the step size are removed.

In this paper we analyzed Algorithm 1 under virtually no restriction on the step size parameter. For the sake of simplicity of analysis, we assume the step size parameter is reduced and increased by the same multiplicative factor. This can be relaxed to some degree. We also do not assume that $g(x, \xi')$ is Lipschitz continuous, we only impose this condition on $\phi$. The cost of relaxing all these assumptions is the use of $\epsilon_f$. For simplicity of the analysis, $\epsilon_f$ is assumed to be fixed throughout the algorithm. In practice, it can be re-estimated regularly. In many applications, $\epsilon_f$ tends to get smaller as the algorithm progresses towards optimality. Our experiments show that estimating $\epsilon_f$ is easy and works well in practice. Moreover, one can use much smaller values for $\epsilon_f$ than theory dictates.

# 8   Acknowledgments

This work was partially supported by NSF Grants TRIPODS 17-40796, NSF Grant CCF 2008434 and DARPA Lagrange award HR-001117S0039. Miaolan Xie was partially supported by a PhD Fellowship provided by MunchRe. Billy Jin was partially supported by NSERC fellowship PGSD3-532673-2019.

The authors are grateful to the anonymous referees for their reviews that helped us improve the paper.

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
