# A Appendix: Oracles

**Proposition 1.** *Let $\hat{e}(x, d) := \ell(x, d) - \phi(x)$ be a $(\hat{\nu}(x), \hat{b}(x))$-subexponential random variable and $Var_{d \sim \mathcal{D}} [\ell(x, d)] \leq \hat{\epsilon}(x)^2$, for some $\hat{\nu}(x), \hat{b}(x), \hat{\epsilon}(x)$. Let $e(x, \mathcal{S}) = |f(x, \mathcal{S}) - \phi(x)|$ and $N = |\mathcal{S}|$, then*

$$\mathbb{E}_{\mathcal{S}} \left[ e(x, \mathcal{S}) \right] \leq \frac{1}{\sqrt{N}} \hat{\epsilon}(x) \quad and \quad e(x, \mathcal{S}) \text{ is } (\nu(x), b(x))\text{-subexponential,}$$

*with $\nu(x) = b(x) = 8e^2 \max \left\{ \frac{\hat{\nu}(x)}{\sqrt{N}}, \hat{b}(x) \right\}$.*

**Proof of the Proposition 1**

*Proof.* We will first show that $\mathbb{E}_{\mathcal{S}} e(x, \mathcal{S}) \leq \frac{1}{\sqrt{N}} \hat{\epsilon}(x)$. Since $|\mathcal{S}| = N$, we have

$$\mathbb{E}_{\mathcal{S}} \, e(x, \mathcal{S})^2 = \mathbb{E}_{\mathcal{S}} \left| \frac{1}{N} \sum_{d \in \mathcal{S}} \ell(x, d) - \phi(x) \right|^2$$

$$= \frac{1}{N^2} \cdot \mathbb{E}_{\mathcal{S}} \left| \sum_{d \in \mathcal{S}} (\ell(x, d) - \phi(x)) \right|^2$$

$$= \frac{1}{N} \cdot \mathbb{E}_{d \sim D} |\ell(x, d) - \phi(x)|^2$$

$$\leq \frac{1}{N} \cdot \hat{\epsilon}(x)^2.$$

Combining this with the inequality $[\mathbb{E} \, e(x, S)]^2 \leq \mathbb{E} \left[ e(x, S)^2 \right]$ gives $\mathbb{E}(e(x, S)) \leq \frac{1}{\sqrt{N}} \hat{\epsilon}(x)$.

Next, we will show that $e(x, \mathcal{S})$ is sub-exponential. We have

$$e(x, \mathcal{S}) = \left| \frac{1}{N} \sum_{d \in \mathcal{S}} \ell(x, d) - \phi(x) \right| = \left| \underbrace{\frac{1}{N} \sum_{d \in \mathcal{S}} (\ell(x, d) - \phi(x))}_{Y} \right|.$$

Note that $Y := \frac{1}{N} \sum_{d \in \mathcal{S}} (\ell(x, d) - \phi(x))$ is sub-exponential with parameters $(\frac{\hat{\nu}(x)}{\sqrt{N}}, \hat{b}(x))$, because it is the average of $N$ independent sub-exponential random variables each with parameters $(\hat{\nu}(x), \hat{b}(x))$. For any random variable $Y$, let $\|Y\|_{\psi_1} := \sup_{k \geq 1} \frac{1}{k} \left( \mathbb{E} |Y|^k \right)^{\frac{1}{k}}$ denote the sub-exponential norm of $Y$. Then

$$\|e(x, \mathcal{S}) - \mathbb{E} \, e(x, \mathcal{S})\|_{\psi_1} \leq \|e(x, \mathcal{S})\|_{\psi_1} + \|\mathbb{E} e(x, \mathcal{S})\|_{\psi_1} \quad \text{(triangle inequality)}$$

$$= \|e(x, \mathcal{S})\|_{\psi_1} + |\mathbb{E} e(x, \mathcal{S})| \quad (\|a\|_{\psi_1} = |a| \text{ for any } a \in \mathbb{R})$$

$$\leq \|e(x, \mathcal{S})\|_{\psi_1} + \mathbb{E} |e(x, \mathcal{S})| \quad \text{(Jensen's inequality)}$$

$$\leq 2 \|e(x, \mathcal{S})\|_{\psi_1} \quad (\text{definition of } \|\cdot\|_{\psi_1})$$

$$= 2 \|\|Y\|\|_{\psi_1} \quad (\text{definition of } Y)$$

$$= 2 \|Y\|_{\psi_1} \quad (\text{definition of } \|\cdot\|_{\psi_1})$$

$$\leq 4 \max \left\{ \frac{\hat{\nu}(x)}{\sqrt{N}}, \hat{b}(x) \right\} \quad (\text{Proposition 2.7.1 (e)} \rightarrow \text{(b) in [Ver18]})$$

Applying Proposition 2.7.1 (b) $\rightarrow$ (e) in [Ver18], we get that $e(x, \mathcal{S})$ is $(\hat{m}(x), \hat{m}(x))$-subexponential, where $\hat{m}(x) = 8e^2 \max \left\{ \frac{\hat{\nu}(x)}{\sqrt{N}}, \hat{b}(x) \right\}$.

$\square$

**Proposition 2.** *Let $g = g(x, \mathcal{S})$. Assuming $\mathbb{E}_{d \sim \mathcal{D}} \nabla \ell(x, d) = \nabla \phi(x)$, then*

$$|\mathcal{S}| \geq \frac{M_c + M_v \|\nabla\phi(x)\|^2}{\delta} \min\left\{ \frac{1}{\epsilon_g^2}, \frac{(1 + \kappa\alpha)^2}{\kappa^2 \alpha^2 \|\nabla\phi(x)\|^2} \right\}$$

*implies*

$$\mathbb{P}\left(\|g - \nabla\phi(x)\| \leq \max\{\epsilon_g, \kappa\alpha\|g\|\}\right) \geq 1 - \delta.$$

*This bound implies a looser bound of:*

$$|\mathcal{S}| \geq \max\left\{ \frac{2M_c}{\delta\epsilon_g^2}, \frac{2M_v(1 + \kappa\alpha)^2}{\delta\kappa^2\alpha^2} \right\}.$$

**Proof of Proposition 2**

*Proof.* For any fixed $M > 0$, by Markov inequality, we have

$$\mathbb{P}\left( \left\| \frac{1}{|\mathcal{S}|} \sum_{d \in \mathcal{S}} \nabla\ell(x, d) - \mathbb{E}_{d \sim \mathcal{D}} \nabla\ell(x, d) \right\| > M \right) \leq \frac{\mathbb{E}\left\| \frac{1}{|\mathcal{S}|} \sum_{d \in \mathcal{S}} \nabla\ell(x, d) - \mathbb{E}_{d \sim \mathcal{D}} \nabla\ell(x, d) \right\|^2}{M^2}$$

$$\leq \frac{M_c + M_v \|\nabla\phi(x)\|^2}{|\mathcal{S}| M^2}$$

Let $M = \max\{\epsilon_g, \eta \|\nabla\phi(x)\|\}$, with $\eta = \frac{\kappa\alpha}{1 + \kappa\alpha} \in (0, 1)$. Since $g = \frac{1}{|\mathcal{S}|} \sum_{d \in \mathcal{S}} \nabla\ell(x, d)$, we have

$$\mathbb{P}\left(\|g - \nabla\phi(x)\| \leq \max\{\epsilon_g, \eta \|\nabla\phi(x)\|\}\right) \geq 1 - \frac{M_c + M_v \|\nabla\phi(x)\|^2}{|\mathcal{S}| \cdot \max\{\epsilon_g, \eta \|\nabla\phi(x)\|\}^2}.$$

Notice if $\epsilon_g \geq \eta \|\nabla\phi(x)\|$, the above inequality implies that for $|\mathcal{S}| \geq \frac{M_c + M_v \|\nabla\phi(x)\|^2}{\delta\epsilon_g^2}$, the desired result is obtained. On the other hand, if $\|g - \nabla\phi(x)\| \leq \eta \|\nabla\phi(x)\|$, by triangle inequality:

$$\|\nabla\phi(x)\| - \|g\| \leq \eta \|\nabla\phi(x)\| \implies \eta \|\nabla\phi(x)\| \leq \frac{\eta}{1 - \eta} \|g\|$$

$$\implies \|g - \nabla\phi(x)\| \leq \frac{\eta}{1 - \eta} \|g\| = \kappa\alpha \|g\|.$$

Hence, by picking $|\mathcal{S}| \geq \frac{M_c + M_v \|\nabla\phi(x)\|^2}{\delta} \frac{(1 + \kappa\alpha)^2}{\kappa^2\alpha^2 \|\nabla\phi(x)\|^2}$, we obtain the desired result for the case $\|g - \nabla\phi(x)\| \leq \eta \|\nabla\phi(x)\|$. Putting these two cases together, we get that as long as

$$|\mathcal{S}| \geq \frac{M_c + M_v \|\nabla\phi(x)\|^2}{\delta} \min\left\{ \frac{1}{\epsilon_g^2}, \frac{(1 + \kappa\alpha)^2}{\kappa^2\alpha^2 \|\nabla\phi(x)\|^2} \right\},$$

$\mathbb{P}\left(\|g - \nabla\phi(x)\| \leq \max\{\epsilon_g, \kappa\alpha\|g\|\}\right) \geq 1 - \delta$ will hold.

For simplicity in presentation, this also implies the result holds with

$$|\mathcal{S}| \geq \max\left\{ \frac{2M_c}{\delta\epsilon_g^2}, \frac{2M_v(1 + \kappa\alpha)^2}{\delta\kappa^2\alpha^2} \right\}.$$

$\square$

**Proposition 3.** *Let $g = g(x, \mathcal{U})$, and fix $\epsilon_g = 2\left(\sqrt{n}L\sigma + \frac{\sqrt{n}\epsilon_f}{\sigma}\right)$ where $n$ is the dimension of $x$. Then*

$$|\mathcal{U}| \geq \frac{\frac{3}{4}L^2\sigma^2 n(n+2)(n+4) + \frac{12\epsilon_f^2}{\sigma^2}n + 18n\|\nabla\phi(x)\|^2}{\delta} \min\left\{ \frac{4}{\epsilon_g^2}, \frac{1}{\left(\frac{\kappa\alpha}{1 + \kappa\alpha}\|\nabla\phi(x)\| - \frac{\epsilon_g}{2}\right)^2} \right\}$$

*implies*

$$\mathbb{P}\left(\|g - \nabla\phi(x)\| \leq \max\{\epsilon_g, \kappa\alpha\|g\|\}\right) \geq 1 - \delta.$$

*Note that in the setting, $\epsilon_g$ is a fixed bias dependent on $\sigma$, and cannot be made arbitrarily small.*

**Proof of Proposition 3**

*Proof.* Let $F(x) = \mathbb{E}_{u \sim N(0,I)}[\phi(x + \sigma u)]$ be the Gaussian smoothing of $\phi$, and $N = |\mathcal{U}|$. In Section 2.3 of [BCCS21], it is shown that

$$\|\nabla F(x) - \nabla \phi(x)\| \leq \sqrt{n} L \sigma + \frac{\sqrt{n} \epsilon_f}{\sigma} = \frac{\epsilon_g}{2}, \tag{7}$$

and $g(x, \mathcal{U})$ is an unbiased estimator of $\nabla F(x)$ with $\mathrm{Var}\{g(x, \mathcal{U})\} \preceq \frac{1}{N} \kappa(x) I$ where

$$\kappa(x) = 3 \left( 3 \|\nabla \phi(x)\|^2 + \frac{L^2 \sigma^2}{4} (n+2)(n+4) + \frac{4\epsilon_f^2}{\sigma^2} \right).$$

Note that since $\mathbb{E}[g(x, \mathcal{U})] = \nabla F(x)$, we have

$$\mathbb{E}[\|g(x, \mathcal{U}) - \nabla F(x)\|^2] = \mathrm{tr}(\mathrm{Var}\{g(x, \mathcal{U})\}) \leq \mathrm{tr}\left( \frac{1}{N} \kappa(x) I \right) = \frac{n}{N} \kappa(x).$$

We use these facts to show that Gaussian smoothed gradients gives a valid first order oracle. First, by the triangle inequality, we have

$$\|g(x, \mathcal{U}) - \nabla \phi(x)\| \leq \|g(x, \mathcal{U}) - \nabla F(x)\| + \|\nabla F(x) - \nabla \phi(x)\|.$$

Let $M = \max\{\epsilon_g, \eta \|\nabla \phi(x)\|\}$, where $\eta = \frac{\kappa \alpha}{1 + \kappa \alpha}$. Then we have

$$\begin{aligned}
\mathbb{P}\left( \|g(x, \mathcal{U}) - \nabla \phi(x)\| > M \right) &\leq \mathbb{P}\left( \|g(x, \mathcal{U}) - \nabla F(x)\| + \|\nabla F(x) - \nabla \phi(x)\| > M \right) \\
&\leq \mathbb{P}\left( \|g(x, \mathcal{U}) - \nabla F(x)\| + \frac{\epsilon_g}{2} > M \right) \\
&= \mathbb{P}\left( \|g(x, \mathcal{U}) - \nabla F(x)\| > M - \frac{\epsilon_g}{2} \right) \\
&= \mathbb{P}\left( \|g(x, \mathcal{U}) - \nabla F(x)\|^2 > \left( M - \frac{\epsilon_g}{2} \right)^2 \right) \\
&\leq \frac{\mathbb{E}[\|g(x, \mathcal{U}) - \nabla F(x)\|^2]}{\left( M - \frac{\epsilon_g}{2} \right)^2} \quad \text{(Markov's inequality)} \\
&\leq \frac{n \kappa(x)}{N \left( M - \frac{\epsilon_g}{2} \right)^2}.
\end{aligned}$$

Therefore, if

$$N \geq \frac{n \kappa(x)}{\delta \left( M - \frac{\epsilon_g}{2} \right)^2} = \frac{n \kappa(x)}{\delta} \cdot \min\left\{ \frac{4}{\epsilon_g^2}, \left( \frac{1}{\left( \eta \|\nabla \phi(x)\| - \frac{\epsilon_g}{2} \right)^+} \right)^2 \right\} \tag{8}$$

$$= \frac{n \kappa(x)}{\delta} \cdot \min\left\{ \frac{4}{\epsilon_g^2}, \frac{1}{\left( \eta \|\nabla \phi(x)\| - \frac{\epsilon_g}{2} \right)^2} \right\}, \tag{9}$$

then

$$\mathbb{P}\left( \|g(x, \mathcal{U}) - \nabla \phi(x)\| > \max\{\epsilon_g, \eta \|\nabla \phi(x)\|\} \right) \leq \delta.$$

(To go from (8) to (9), note that when $\eta \|\nabla \phi(x)\| - \frac{\epsilon_g}{2}$ is negative, it is greater than $-\frac{\epsilon_g}{2}$.)

Now, using the same argument as in the proof of Proposition 2, we get that

$$\|g(x, \mathcal{U}) - \nabla \phi(x)\| > \max\{\epsilon_g, \kappa \alpha \|g(x, \mathcal{U})\|\} \quad \text{implies} \quad \|g(x, \mathcal{U}) - \nabla \phi(x)\| > \max\{\epsilon_g, \eta \|\nabla \phi(x)\|\}.$$

Therefore, if $N$ is greater than or equal to the bound in (9), then

$$\mathbb{P}\left( \|g(x, \mathcal{U}) - \nabla \phi(x)\| > \max\{\epsilon_g, \kappa \alpha \|g(x, \mathcal{U})\|\} \right) \leq \delta.$$

$\square$

# B Appendix: Proof of Lemma 2

**Lemma 2.** *For any positive integer $t$ and any $\hat{p} \in (\frac{1}{2}, 1]$, we have*

$$\mathbb{P}\left(T_\varepsilon > t \text{ and } \sum_{k=0}^{t-1} I_k \geq \hat{p}t \text{ and } \sum_{k=0}^{t-1} U_k \Theta_k I_k < \left(\hat{p} - \frac{1}{2}\right)t - \frac{d}{2}\right) = 0,$$

*where $d = \max\left\{-\frac{\ln \alpha_0 - \ln \bar{\alpha}}{\ln \gamma}, 0\right\}$.*

To prove Lemma 2, we will first prove two additional lemmas. The first lemma shows that the number of large and successful iterations is bounded below by the number of large and unsuccessful ones up to a constant.

**Lemma 3.** *Let $d = \max\left\{-\frac{\ln \alpha_0 - \ln \bar{\alpha}}{\ln \gamma}, 0\right\}$. For any positive integer $t$, we have*

$$\sum_{k=0}^{t-1} U_k \Theta_k \geq \sum_{k=0}^{t-1} U_k(1 - \Theta_k) - d.$$

*Proof.* The proof follows simply from the fact that any unsuccessful step decreases the step size by a factor of $\gamma$, while any large successful step increases the step by a factor of $\gamma^{-1}$. Since a large step at iteration $k$ has both $\alpha_k$ and $\alpha_{k+1}$ bounded from below by $\bar{\alpha}$, every time $\alpha_k$ gets decreased has to correspond to a large step where it gets increased, except for at most $\max\{-(\ln \alpha_0 - \ln \bar{\alpha})/\ln \gamma, 0\}$ iterations, which is the number of unsuccessful steps it takes to decrease step size from $\alpha_0$ to $\bar{\alpha}$. $\square$

(Without loss of generality, one may assume $\alpha_0 \geq \bar{\alpha}$, as $\alpha_0$ can chosen to be large.)

**Corollary 1.** *From Lemma 3 we have*

$$\sum_{k=0}^{t-1} U_k \Theta_k \geq \frac{1}{2}\left(\sum_{k=0}^{t-1} U_k - d\right).$$

The next Lemma is an analogue of Lemma 3 for the *small* steps, it states that the number of small true steps is upper-bounded by the number of small false steps.

**Lemma 4.** *For any positive integer $t < T_\varepsilon$, we have:*

$$\sum_{k=0}^{t-1} (1 - U_k)I_k \leq \sum_{k=0}^{t-1} (1 - U_k)(1 - I_k).$$

*Proof.* We have

$$\sum_{k=0}^{t-1} (1 - U_k)I_k \leq \sum_{k=0}^{t-1} (1 - U_k)\Theta_k \leq \sum_{k=0}^{t-1} (1 - U_k)(1 - \Theta_k) \leq \sum_{k=0}^{t-1} (1 - U_k)(1 - I_k).$$

The first inequality follows from Assumption 3(iv), which implies that the number of small successful iterations is at least the number of small true iterations. The second inequality follows from the fact that the number of small steps where $\alpha_k$ is increased is bounded by the number of small steps where $\alpha_k$ is decreased. The third inequality again uses Assumption 3 (iv), since any small unsuccessful has to be false. $\square$

We are now ready to prove Lemma 2.

*For any positive integer $t$ and any $\hat{p} \in (\frac{1}{2}, 1]$, we have*

$$\mathbb{P}\left(T_\varepsilon > t \text{ and } \sum_{k=0}^{t-1} I_k \geq \hat{p}t \text{ and } \sum_{k=0}^{t-1} U_k \Theta_k I_k < \left(\hat{p} - \frac{1}{2}\right)t - \frac{d}{2}\right) = 0.$$

*Proof.* It suffices to show that the two events $T_\varepsilon > t$ and $\sum_{k=0}^{t-1} I_k \geq \hat{p}t$ together imply $\sum_{k=0}^{t-1} U_k \Theta_k I_k \geq \left(\hat{p} - \frac{1}{2}\right) t - \frac{d}{2}$. In the remainder of the proof, assume that $T_\varepsilon > t$ and $\sum_{k=0}^{t-1} I_k \geq \hat{p}t$.

Among the first $t$ steps, let

- $L_t = \sum_{k=0}^{t-1} U_k I_k$ be the number of true large steps,

- $L_f = \sum_{k=0}^{t-1} U_k(1 - I_k)$ be the number of false large steps,

- $S_t = \sum_{k=0}^{t-1}(1 - U_k)I_k$ be the number of true small steps,

- $S_f = \sum_{k=0}^{t-1}(1 - U_k)(1 - I_k)$ be the number of false small steps,

- $L = L_t + L_f$ be the number of large steps,

- $S = S_t + S_f$ be the number of small steps.

Observe that $L + S = t$, because every step is either large or small. Moreover, since $\sum_{k=0}^{t-1} I_k \geq \hat{p}t$, this implies

$$L_f + S_f = \sum_{k=0}^{t-1}(1 - I_k) \leq n - \hat{p}t. \tag{10}$$

Also, from Lemma 4 and the fact that $n < T_\epsilon$, we know that

$$S_t = \sum_{k=0}^{t-1}(1 - U_k)I_k \leq \sum_{k=0}^{t-1}(1 - U_k)(1 - I_k) = S_f \tag{11}$$

Now, recall from Corollary 1 that the number of large, successful steps is $\sum_{k=0}^{t-1} U_k \Theta_k \geq \frac{1}{2}(L - d)$. Also, note that

$$\sum_{k=0}^{t-1} U_k \Theta_k = \sum_{k=0}^{t-1} U_k \Theta_k I_k + \sum_{k=0}^{t-1} U_k \Theta_k (1 - I_k).$$

This implies that the number of large, successful, true steps is at least

$$\sum_{k=0}^{t-1} U_k \Theta_k I_k \geq \frac{L}{2} - \frac{d}{2} - \sum_{k=0}^{t-1} U_k \Theta_k (1 - I_k)$$

$$\geq \frac{L}{2} - \frac{d}{2} - \sum_{k=0}^{t-1} U_k(1 - I_k)$$

$$= \frac{L}{2} - \frac{d}{2} - L_f$$

$$= \frac{t - S_t - S_f}{2} - \frac{d}{2} - L_f$$

$$\geq \frac{t - S_t - S_f}{2} - \frac{d}{2} - ((1 - \hat{p})t - S_f) \qquad \text{(by 10)}$$

$$= \frac{S_f - S_t}{2} + \left(\hat{p} - \frac{1}{2}\right)t - \frac{d}{2}$$

$$\geq \left(\hat{p} - \frac{1}{2}\right)t - \frac{d}{2} \qquad \text{(by 11)}$$

$\square$

## C   Appendix: Proof of Theorem 2

**Theorem 2** (Iteration complexity in the bounded noise setting). *Suppose Assumption 3 holds, and $e_k, e_k^+ \leq \epsilon_f$ at every iteration. Then for any $\hat{p} \in (\frac{1}{2} + \frac{4\epsilon_f}{h(\bar{\alpha})}, p)$, and $t \geq \frac{R}{\hat{p} - \frac{1}{2} - \frac{4\epsilon_f}{h(\bar{\alpha})}}$ we have*

$$\mathbb{P}\left(T_\varepsilon \leq t\right) \geq 1 - \exp\left(-\frac{(p - \hat{p})^2}{2p^2}t\right),$$

*where $R = \frac{Z_0}{h(\bar{\alpha})} + \frac{d}{2}$ and $d = \max\left\{-\frac{\ln \alpha_0 - \ln \bar{\alpha}}{\ln \gamma}, 0\right\}$.*

*Proof.* In the bounded noise case, Assumption 3 tells us that as long as $k < T_\varepsilon$, we have $Z_{k+1} \leq Z_k - h(\bar{\alpha}) + 4\epsilon_f$ if $U_k I_k \Theta_k = 1$, and $Z_{k+1} \leq Z_k + 4\epsilon_f$ if $U_k I_k \Theta_i = 0$.

The event $T_\varepsilon > t$ implies that $Z_t > 0$ (since $Z_t = 0$ can only happen at global optimality, hence $T_\varepsilon \leq t$), this in turn implies the event $\sum_{k=0}^{t-1} U_k I_k \Theta_k < (\hat{p} - \frac{1}{2})t - \frac{d}{2}$. To see this, assume that $\sum_{k=0}^{t-1} U_k I_k \Theta_k \geq (\hat{p} - \frac{1}{2})t - \frac{d}{2}$, then

$$Z_t \leq Z_0 - \left[\left(\left(\hat{p} - \frac{1}{2}\right) \cdot t - \frac{d}{2}\right)h(\bar{\alpha}) - t \cdot 4\epsilon_f\right] = Z_0 - \left(\left(\hat{p} - \frac{1}{2}\right)h(\bar{\alpha}) - 4\epsilon_f\right)t + \frac{d}{2} \cdot h(\bar{\alpha}) \leq 0.$$

The last inequality above used the assumptions that $\hat{p} \geq \frac{1}{2} + \frac{4\epsilon_f}{h(\bar{\alpha})}$ and $t \geq \frac{R}{\hat{p} - \frac{1}{2} - \frac{4\epsilon_f}{h(\bar{\alpha})}}$.

Thus, we get

$$\mathbb{P}(T_\varepsilon > t) = \mathbb{P}\left(T_\varepsilon > t, \sum_{i=0}^{t-1} U_i \Theta_i I_i < \left(\hat{p} - \frac{1}{2}\right)t - \frac{d}{2}\right)$$

$$= \mathbb{P}\left(T_\varepsilon > t, \sum_{k=0}^{t-1} U_k \Theta_k I_k < \left(\hat{p} - \frac{1}{2}\right)t - \frac{d}{2}, \sum_{k=0}^{t-1} I_k < \hat{p}t\right)$$

$$+ \mathbb{P}\left(T_\varepsilon > t, \sum_{k=0}^{t-1} U_k \Theta_k I_k < \left(\hat{p} - \frac{1}{2}\right)t - \frac{d}{2}, \sum_{k=0}^{t-1} I_k \geq \hat{p}t\right)$$

$$\leq \mathbb{P}\left(\sum_{k=0}^{t-1} I_k < \hat{p}t\right) + 0 \leq \exp\left(-\frac{(p - \hat{p})^2}{2p^2}t\right).$$

Here, the first equality is due to the fact that the event $T_\varepsilon > t$ implies the event $\sum_{k=0}^{t-1} U_k \Theta_k I_k < \left(\hat{p} - \frac{1}{2}\right)t - \frac{d}{2}$. The first inequality uses Lemma 2, and the last inequality is by Lemma 1. □

## D   Appendix: Proof of Theorem 3

**Theorem 3** (Iteration complexity in the sub-exponential noise setting). *Suppose Assumptions 2 and 3 hold. Then for any $s \geq 0$, $\hat{p} \in (\frac{1}{2} + \frac{4\epsilon_f + s}{h(\bar{\alpha})}, p)$, and $t \geq \frac{R}{\hat{p} - \frac{1}{2} - \frac{4\epsilon_f + s}{h(\bar{\alpha})}}$, we have*

$$\mathbb{P}\left(T_\varepsilon \leq t\right) \geq 1 - \exp\left(-\frac{(p - \hat{p})^2}{2p^2}t\right) - e^{-\min\left\{\frac{s^2 t}{8\nu^2}, \frac{st}{4b}\right\}},$$

*where $R = \frac{Z_0}{h(\bar{\alpha})} + \frac{d}{2}$ and $d = \max\left\{-\frac{\ln \alpha_0 - \ln \bar{\alpha}}{\ln \gamma}, 0\right\}$.*

*Proof.* By Assumption 3, for all $k < T_\varepsilon$, we have $Z_{k+1} \leq Z_k - h(\bar{\alpha}) + 2\epsilon_f + e_k + e_k^+$ if $U_k I_k \Theta_k = 1$, and $Z_{k+1} \leq Z_k + 2\epsilon_f + e_k + e_k^+$ if $U_k I_k \Theta_k = 0$. By the definition of the zeroth order oracle (1),

we know that $\mathbb{E}[e_k]$ and $\mathbb{E}[e_k^+]$ are bounded above by $\epsilon_f$ for all $k$. By the law of total probability,

$$
\mathbb{P}\left(T_\varepsilon > t\right) = \mathbb{P}\underbrace{\left(T_\varepsilon > t, \frac{1}{t}\sum_{k=0}^{t-1}(2\epsilon_f + e_k + e_k^+) \le 4\epsilon_f + s\right)}_{A}
$$

$$
+ \mathbb{P}\underbrace{\left(T_\varepsilon > t, \frac{1}{t}\sum_{k=0}^{t-1}(2\epsilon_f + e_k + e_k^+) > 4\epsilon_f + s\right)}_{B}
$$

First we bound $\mathbb{P}(B)$. For each $k$, since $e_k$ and $e_k^+$ satisfy the one-sided sub-exponential bound 1 with parameters $(\nu, b)$, it is not hard to show that $e_k + e_k^+$ satisfy 1 with parameters $(2\nu, 2b)$. Moreover, since $e_k + e_k^+$ has mean bounded by $2\epsilon_f$, applying (one-sided) Bernstein's inequality, gives for any $s \ge 0$:

$$
\mathbb{P}(B) \le \mathbb{P}\left(\frac{1}{t}\sum_{k=0}^{t-1}\left(e_k + e_k^+\right) > 2\epsilon_f + s\right) \le e^{-\min\{\frac{s^2 t}{8\nu^2}, \frac{st}{4b}\}}.
$$

To bound $\mathbb{P}(A)$ we apply the law of total probability again,

$$
\mathbb{P}(A) = \mathbb{P}\underbrace{\left(T_\varepsilon > t, \frac{1}{t}\sum_{k=0}^{t-1}(2\epsilon_f + e_k + e_k^+) \le 4\epsilon_f + s, \sum_{k=0}^{t-1}\Theta_k I_k U_k < \left(\hat{p} - \frac{1}{2}\right)t - \frac{d}{2}\right)}_{A_1}
$$

$$
+ \mathbb{P}\underbrace{\left(T_\varepsilon > t, \frac{1}{t}\sum_{k=0}^{t-1}(2\epsilon_f + e_k + e_k^+) \le 4\epsilon_f + s, \sum_{k=0}^{t-1}\Theta_k I_k U_k \ge \left(\hat{p} - \frac{1}{2}\right)t - \frac{d}{2}\right)}_{A_2}
$$

Using the same logic as the first parts of the proof of Theorem 2 we show that $P(A_2) = 0$ since $T_\varepsilon > t$ and $\frac{1}{t}\sum_{k=0}^{t-1}(2\epsilon_f + e_k + e_k^+) \le 4\epsilon_f + s$ together imply that $\sum_{k=0}^{t-1}\Theta_k I_k U_k < \left(\hat{p} - \frac{1}{2}\right)t - \frac{d}{2}$. Then, by the second part of the proof of Theorem 2 we have

$$
\mathbb{P}(A_1) \le \mathbb{P}\left(T_\varepsilon > t, \sum_{k=0}^{t-1}\Theta_k I_k U_k < \left(\hat{p} - \frac{1}{2}\right)t - \frac{d}{2}\right)
$$

$$
\le \exp\left(-\frac{(p - \hat{p})^2}{2p^2}t\right).
$$

Combining $\mathbb{P}(A)$ and $\mathbb{P}(B)$, we conclude the proof.

$\square$

# E  Appendix: Assumption 3 holds for Algorithm 1

In this section, we verify that Assumption 3 holds for Algorithm 1 when applied to smooth functions which will allow us to apply the results in Section 4 to derive a high-probability bound on complexity.

As noted earlier, when either $\epsilon_f$ or $\epsilon_g$ are not zero, Algorithm 1 does not converge to a stationary point, but converges to a neighborhood where $\|\nabla\phi(x)\| \le \varepsilon$, with $\varepsilon$ bounded from below in terms of $\epsilon_f$ or $\epsilon_g$. The specific relationship is as follows.

**Inequality 1** (Lower bound on $\varepsilon$).

$$\varepsilon > \max\left\{\frac{\epsilon_g}{\eta}, \max\left\{1 + \kappa\alpha_{\max}, \frac{1}{1-\eta}\right\} \cdot \sqrt{\frac{4\epsilon_f}{\theta(p-\frac{1}{2})} \cdot \max\left\{\frac{0.5L+\kappa}{1-\theta}, \frac{L(1-\eta)}{2(1-2\eta-\theta(1-\eta))}\right\}}\right\},$$

*for some $\eta \in (0, \frac{1-\theta}{2-\theta})$.*

Notice this bound is slightly more general than the one in the main body. If one assumes the oracle can be made so that $p$ is arbitrarily close to 1, then we have the following bound as in Assumption 4 in the main body:

$$\varepsilon > \max\left\{\frac{\epsilon_g}{\eta}, \max\left\{1 + \kappa\alpha_{\max}, \frac{1}{1-\eta}\right\} \cdot \sqrt{\frac{8\epsilon_f}{\theta} \cdot \max\left\{\frac{0.5L+\kappa}{1-\theta}, \frac{L(1-\eta)}{2(1-2\eta-\theta(1-\eta))}\right\}}\right\}.$$

We restate Assumption 3 below for convenience.

**Assumption 3** (Properties of the stochastic process). *There exist a constant $\bar{\alpha} > 0$ and a non-decreasing function $h : \mathbb{R} \to \mathbb{R}$, which satisfies $h(\alpha) > 0$ for any $\alpha > 0$, such that for any realization of the algorithm, the following hold for all $k < T_\varepsilon$:*

(i) $h(\bar{\alpha}) > 8\epsilon_f$.

(ii) $\mathbb{P}(I_k = 1 \mid \mathcal{F}_{k-1}) \geq p$ *for all $k$, with some $p \in (\frac{1}{2} + \frac{4\epsilon_f}{h(\bar{\alpha})}, 1]$.*

(iii) *If $I_k\Theta_k = 1$ then $Z_{k+1} \leq Z_k - h(\alpha_k) + 4\epsilon_f$. (True, successful iterations make progress.)*

(iv) *If $\alpha_k \leq \bar{\alpha}$ and $I_k = 1$ then $\Theta_k = 1$.*

(v) $Z_{k+1} \leq Z_k + 2\epsilon_f + e_k + e_k^+$ *for all $k$.*

**Proposition 4** (Assumption 3 holds for Algorithm 1). *If Inequality 1 and Assumption 1 and 2 hold, then Assumption 3 holds for Algorithm 1 with the following $p$, $\bar{\alpha}$ and $h(\alpha)$:*

1. $p = 1 - \delta$ *when the noise is bounded by $\epsilon_f$, and $p = 1 - \delta - \exp\left(-\min\{\frac{u^2}{2\nu^2}, \frac{u}{2b}\}\right)$ otherwise. Here $u = \inf_x\{\epsilon_f - \mathbb{E}[e(x)]\}$.*

2. $\bar{\alpha} = \min\left\{\frac{1-\theta}{0.5L+\kappa}, \frac{2(1-2\eta-\theta(1-\eta))}{L(1-\eta)}\right\}$.

3. $h(\alpha) = \min\left\{\frac{\theta\epsilon^2\alpha}{(1+\kappa\alpha_{\max})^2}, \theta\alpha(1-\eta)^2\epsilon^2\right\}$.

*Proof.* We will show that each item in Assumption 3 holds. Throughout the proof, we use $f(x)$ to mean $f(x, \xi(x))$ for clarity.

**(i)** We need to show that $h(\bar{\alpha}) > 8\epsilon_f$. Indeed, using

$$h(\bar{\alpha}) = \min\left\{\frac{1}{(1+\kappa\alpha_{\max})^2}, (1-\eta)^2\right\}\theta\varepsilon^2\bar{\alpha},$$

with

$$\bar{\alpha} = \min\left\{\frac{1-\theta}{0.5L+\kappa}, \frac{2(1-2\eta-\theta(1-\eta))}{L(1-\eta)}\right\},$$

and the Inequality 1 on $\varepsilon$, we conclude that (i) holds.

**(ii)** We denote $J_k := \mathbb{1}\{\|g_k - \nabla\phi(x_k)\| \leq \max\{\epsilon_g, \kappa\mathcal{A}_k\|g_k\|\}\}$.

Clearly, we have

$$\mathbb{P}(I_k = 0 \mid \mathcal{F}_{k-1}) = \mathbb{P}(J_k = 0 \text{ or } e_k + e_k^+ > 2\epsilon_f \mid \mathcal{F}_{k-1})$$
$$\leq \mathbb{P}(J_k = 0 \mid \mathcal{F}_{k-1}) + \mathbb{P}(e_k + e_k^+ > 2\epsilon_f \mid \mathcal{F}_{k-1})$$

The first term is bounded above by $\delta$, based on the use of the first order oracle. The second term is zero in the case when $\epsilon_f$ is a deterministic bound on the noise. Otherwise,

since $e_k$ and $e_k^+$ individually satisfy the one-sided sub-exponential bound in 1 with parameters $\epsilon_f$ and $(\nu, b)$, it is not hard to show that $e_k + e_k^+$ satisfy 1 with parameters $2\epsilon_f$ and $(2\nu, 2b)$. Hence by (one-sided) Bernstein's inequality, the second term is bounded above by $e^{-\min\{\frac{u^2}{2\nu^2}, \frac{u}{2b}\}}$. (Recall that $u = \inf_x\{\epsilon_f - \mathbb{E}[e(x)]\}$.) Thus we have shown that

$$\mathbb{P}(I_k = 1 \mid \mathcal{F}_{k-1}) \geq p$$

for all $k$, for the $p$ in the statement of this Proposition.

The fact $p \in (\frac{1}{2} + \frac{4\epsilon_f}{h(\bar{\alpha})}, 1]$ follows from the definitions of $h$ and $\bar{\alpha}$ in the statement of this Proposition, together with the Inequality 1 on $\varepsilon$.

**(iii)** Since iteration $k$ is true, we know that $\|g_k - \nabla\phi(x_k)\| \leq \max\{\epsilon_g, \kappa\alpha_k\|g_k\|\}$. We consider two cases:

- Suppose $\|g_k - \nabla\phi(x_k)\| \leq \kappa\alpha_k\|g_k\|$. By triangle inequality, we get

$$\|g_k\| \geq \frac{1}{1 + \kappa\alpha_k}\|\nabla\phi(x_k)\| \geq \frac{1}{1 + \kappa\alpha_{\max}}\|\nabla\phi(x_k)\|.$$

Together with the fact that iteration $k$ is successful, we obtain

$$f(x_{k+1}) - f(x_k) \leq -\alpha_k\theta\|g_k\|^2 + 2\epsilon_f \leq -\frac{\alpha_k\theta\|\nabla\phi(x_k)\|^2}{(1 + \kappa\alpha_{\max})^2} + 2\epsilon_f.$$

- Suppose $\|g_k - \nabla\phi(x_k)\| \leq \epsilon_g$. Since $k < T_\varepsilon$, we have $\|\nabla\phi(x_k)\| > \varepsilon \geq \frac{\epsilon_g}{\eta}$. This implies that $\|g_k - \nabla\phi(x_k)\| \leq \eta\|\nabla\phi(x_k)\|$. Rearranging this using the triangle inequality, we get that

$$\|g_k\| \geq (1 - \eta)\|\nabla\phi(x_k)\|.$$

Putting this together with the fact that iteration $k$ is successful, we obtain

$$f(x_{k+1}) - f(x_k) \leq -\alpha_k\theta\|g_k\|^2 + 2\epsilon_f \leq -\alpha_k\theta(1 - \eta)^2\|\nabla\phi(x_k)\|^2 + 2\epsilon_f.$$

Combining the above two cases, we get that on any true, successful iteration with $k < T_\varepsilon$, the following inequality holds:

$$f(x_{k+1}) - f(x_k) \leq -\min\left\{\frac{1}{(1 + \kappa\alpha_{\max})^2}, (1 - \eta)^2\right\}\alpha_k\theta\|\nabla\phi(x_k)\|^2 + 2\epsilon_f.$$

By $k < T_\varepsilon$, we know $\|\nabla\phi(x_k)\| > \varepsilon$, so the above inequality implies $f(x_{k+1}) - f(x_k) \leq -h(\alpha_k) + 2\epsilon_f$. Finally, because $e_k + e_k^+ \leq 2\epsilon_f$ on true iterations, we get $\phi(x_{k+1}) - \phi(x_k) \leq -h(\alpha_k) + 4\epsilon_f$. Recall that $Z_k = \phi(x_k) - \phi^*$, so $Z_{k+1} - Z_k = \phi(x_{k+1}) - \phi(x_k)$. This proves (iii).

**(iv)** We first show that if $\alpha_k \leq \bar{\alpha}$ and $I_k = 1$, then

$$\phi(x_k - \alpha_k g_k) \leq \phi(x_k) - \alpha_k\theta\|g_k\|^2. \tag{12}$$

Since $I_k = 1$, $\|g_k - \nabla\phi(x_k)\| \leq \max\{\kappa\alpha_k\|g_k\|, \epsilon_g\}$. Just like in the proof of (iii), we consider two cases:

- Suppose $\|g_k - \nabla\phi(x_k)\| \leq \kappa\alpha_k\|g_k\|$. Then since $\alpha_k \leq \bar{\alpha} \leq \frac{1-\theta}{0.5L+\kappa}$, by Assumption 1 and Lemma 3.1 of [CS17], we have 12 hold.
- Suppose $\|g_k - \nabla\phi(x_k)\| \leq \epsilon_g$. Since $k < T_\epsilon$, we have $\|\nabla\phi(x_k)\| > \varepsilon \geq \frac{\epsilon_g}{\eta}$ by Inequality 1. Therefore, $\|g_k - \nabla\phi(x_k)\| \leq \eta\|\nabla\phi(x_k)\|$. Combining this with the fact that $\alpha_k \leq \bar{\alpha} \leq \frac{2(1-2\eta-\theta(1-\eta))}{L(1-\eta)}$, by Assumption 1 and Lemma 4.3 of [BCS19] (applied with $\epsilon_f = 0$), we have 12 hold.

Now, recalling the definitions of $e_k$ and $e_k^+$ and using the fact that $e_k + e_k^+ \leq 2\epsilon_f$ (since $I_k = 1$), inequality (12) implies

$$f(x_k - \alpha_k g_k) \leq f(x_k) - \alpha_k\theta\|g_k\|^2 + e_k + e_k^+ \leq f(x_k) - \alpha_k\theta\|g_k\|^2 + 2\epsilon_f,$$

which proves (iv).

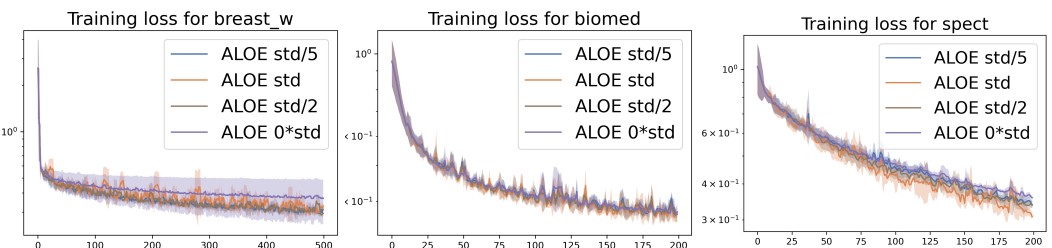

Figure 4: ALOE is robust to the choice of $\epsilon_f$.

**(v)** Note that $Z_{k+1} = Z_k$ on any unsuccessful iteration, so the inequality holds trivially in that case. On the other hand, if iteration $k$ is successful, then by the modified Armijo condition, we have

$$f(x_{k+1}) - f(x_k) \leq -\alpha_k \theta \|g_k\|^2 + 2\epsilon_f \leq 2\epsilon_f.$$

This implies that $\phi(x_{k+1}) - \phi(x_k) \leq 2\epsilon_f + e_k + e_k^+$. Since $Z_{k+1} - Z_k = \phi(x_{k+1}) - \phi(x_k)$, (v) is proved.

$\square$

Applying it to Theorem 3 gives the explicit complexity bound for Algorithm 1:

**Theorem 4.** *Suppose the Inequality 1 on $\varepsilon$ is satisfied for some $\eta \in (0, \frac{1-\theta}{2-\theta})$, and Assumptions 1 and 2 hold, then we have the following bound on the iteration complexity: For any $s \geq 0$, $\hat{p} \in (\frac{1}{2} + \frac{4\epsilon_f + s}{C\varepsilon^2}, p)$, and $t \geq \frac{R}{\hat{p} - \frac{1}{2} - \frac{4\epsilon_f + s}{C\varepsilon^2}}$,*

$$\mathbb{P}\left(T_\varepsilon \leq t\right) \geq 1 - \exp\left(-\frac{(p-\hat{p})^2}{2p^2}t\right) - \exp\left(-\min\left\{\frac{s^2 t}{8\nu^2}, \frac{st}{4b}\right\}\right).$$

*Here, $R = \frac{\phi(x_0) - \phi^*}{C\varepsilon^2} + \max\left\{-\frac{\ln \alpha_0 - \ln \bar{\alpha}}{\ln \gamma}, 0\right\}$, $C = \min\left\{\frac{1}{(1+\kappa\alpha_{\max})^2}, (1-\eta)^2\right\}\bar{\alpha}\theta$, with $p$ and $\bar{\alpha}$ as defined in Proposition 4.*

## F   Appendix: Additional experimental results

We now present additional experimental results with ALOE. The first set of experiments shown in Figure 4 compares the performance of ALOE using different choices of the error threshold $\epsilon_f$. We observe that the algorithm is fairly robust with respect to these choices, but overall selecting $\epsilon_f > 0$ is justified in practice.

Next we show results of applying ALOE (and SLS) on two different neural network architectures, using softmax loss function, trained on MNIST Handwritten Digit Classification Dataset. The first architecture is a multi-layer perceptron (MLP) neural network that has four layers: an input layer with 784 nodes, two hidden layer with 512 and 256 nodes, and an output layer with 10 nodes. All activation functions are ReLU. This is the same architecture as in [VML+19]. The results are shown in Figure 5. The second network is a small convolutional neural network (CNN) that in addition to the input and output layers, has two convolutional layers and one fully connected layer. Each convolutional layer uses a $3 \times 3$ kernel with a stride length of 1, and is followed by a $2 \times 2$ max pooling. This architecture follows the tutorial at this link. The results are in Figure 6.

Thus we have two non-convex problems, to which we apply two versions of ALOE and one version of SLS. The algorithms are: 1) ALOE 1 is the ALOE algorithm with $\gamma = 0.7$, 2) ALOE 2 is the ALOE algorithm with $\gamma = 0.9$. 3) SLS algorithm uses the suggested parameters as in the paper [VML+19]. The reason for using $\gamma = 0.9$ is because in these experiments this value is closer to the parameters chosen by SLS, where $\gamma$ is chosen heuristically, depending on the size of the data set and the mini-batch size.

In each figure, the leftmost plot shows the progress of the *training* loss of each algorithm, the central plot shows the progress of the *testing* loss, and the rightmost plot shows the progress of the validation error.

Figure 5 demonstrates that for the MLP, the behavior of ALOE algorithm on the testing and training loss is somewhat similar - both losses start to increase at about the time when the validation accuracies plateau at about 98%. SLS however, like most SGD algorithms, continues to improve the training loss without improving the validation loss. We find this observation supports our theoretical results that ALOE optimizes the expected loss, rather than the empirical loss, but only up to certain accuracy. The average test set error rates for all three algorithms in this case are similar, which are about 2%. In Figure 6 we observe that ALOE manages to reach much better accuracy than SLS on the CNN. Specifically, the average test set error rates for ALOE 1, ALOE 2 and SLS are 0.9%, 0.9% and 51% respectively. In Figure 7, we plot the step sizes for each algorithm. We hypothesize that the reason SLS does not perform well on this problem is because the step size becomes quite small and the algorithm does not manage to progress. ALOE however, is able to take large steps and progresses well.

Careful exploration of practical variants of ALOE, such as heuristics for $\gamma$, $\epsilon_f$ and a minibatch size are subject for future research.

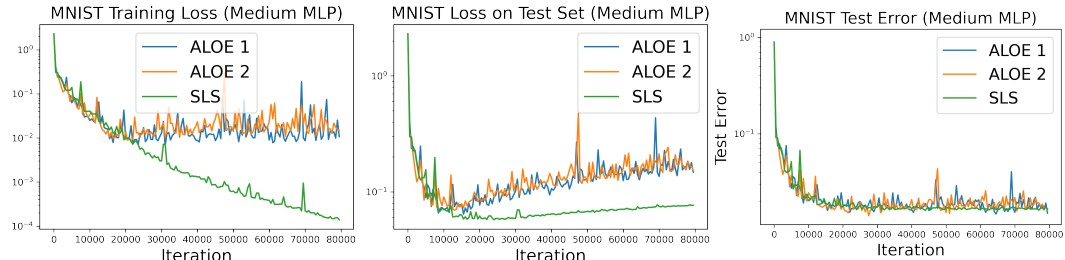

Figure 5: MNIST with multi-layer perceptron neural network

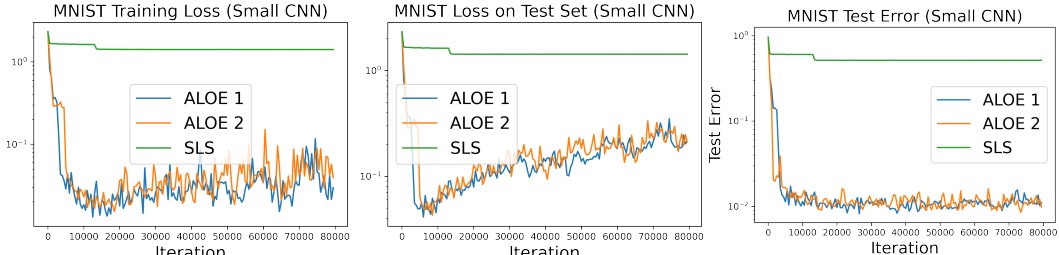

Figure 6: MNIST with CNN

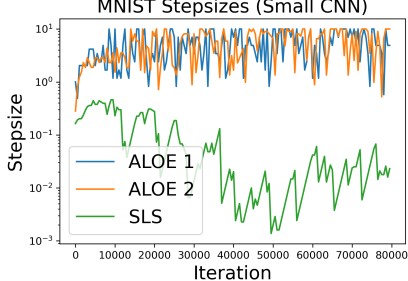

Figure 7: Step-sizes for MNIST with CNN