# OpenReview forum: "High Probability Complexity Bounds for Line Search Based on Stochastic Oracles"
_NeurIPS.cc/2021/Conference — NeurIPS 2021 Poster_

### Official Review · Reviewer_ApuJ · 2021-07-07

**Rating:** 6
**Confidence:** 4

**Summary:**

The paper considers a new line search variant for minimizing functions with access to function value (zero order) and gradient (first order) estimates only through probabilistic oracles.
For both oracles the output is a function of the input point and a random variable, and with probability 1-delta the output is close to the actual value. For the remaining delta probability, the first order oracle assumes nothing while the zero order oracle assumes that large errors are increasingly unlikely, making the requirements for the two oracles different.

The random variable in the oracle, could for instance represent the mini-batch used to estimate the required quantity.
The authors propose a variant of Armijo Line search that
 - Adds some slack on the Armijo condition depending on the quality of the zeroth order oracle
 - Recompute gradient and function values in each step in the line search instead of using the same one.

For this line search method the authors show high probability iteration bounds for minimizing a Lipshitz bounded smooth function,
that converge to an epsilon stationary point (norm gradient less than epsilon) where epsilon is lower bounded by the quality of the oracles.
The bounds are more general in the sense that less is assumed about the oracles, and that the paper shows high probability iteration bounds, where the usual is a bound on the expected number of iterations.


**Limitations And Societal Impact:**

Yes

**Main Review:**

The approach and analysis is made as general as possible and analyzed without assuming convexity. However all the experiments consider a convex problem.
Nonetheless, the experiments confirm that the proposed line searching method works well in this case.
It would have been interesting to see non-convex problem, in particular the case with zeroth order optimization from for instance reinforcement learning as those oracles seem to able to be particularly noisy,
albeit such problems may have a lot of moving parts to control that make it hard to isolate the value of the algorithm proposed.

It seems in the general case minibatch size should depend on the learning rate in the algorithm (in the case where the randomness comes from the minibatches),
at least for the oracles created from mini batches satisfy the assumptions required? Stated differently, if batch size is too small there seems to be no guarantee for the algorithm. What happens in the experiments if the batch size is made much smaller?

Overall I think the proposed algorithm is nice and fairly simple which is positive albeit there are some  parameters that need setting.
The actual Theorem has a lot of parameters but the overall message can be extracted, and the result is quite general.

- The norms in equation 2, and in general, is that any norm? or maybe the two norm of the vector?
- Assumption 2, is that a standard assumption that the errors in different steps are independent (they could depend on the same x here)?
- Does the iteration bound (almost) match an obvious lower bound for this kind of oracles?
- Can it be shown that for these kinds of oracles the recomputation of function value and gradient in each step in the line search is necessary in some sense (could the standard armijo SGD also converge here as that may be cheaper to run (data could be expensive to generate for instance with zeroth order optimization with finite difference approximation in reinforcement learning).
- why use absolute values around |e(x)|  on line 200 - 202, e(x) is already non-negative. Am I missing something there?


**Time Spent Reviewing:**

9

---

> ### Author Response · Authors · 2021-08-10
> **Response to Reviewer ApuJ**
>
> Thank you for the review and questions!
>
> ----
>
> **Question 1**: *It would have been interesting to see non-convex problem, in particular the case with zeroth order optimization from for instance reinforcement learning as those oracles seem to able to be particularly noisy, albeit such problems may have a lot of moving parts to control that make it hard to isolate the value of the algorithm proposed.*
>
> **Response 1**: The supplementary material includes experiments with neural networks, that are non-convex and show even better performance of the proposed method. We agree that it may be better to move them into the main paper. It is indeed a more complex task to find appropriate and illustrative RL examples. A similar line search algorithm has been used for MuJoCo tasks in [BCCS21], but those tasks present a very limited and so far poorly understood class of optimization functions (these functions heavily depend on the policy parametrization).
>
> ----
>
> **Question 2**: *It seems in the general case minibatch size should depend on the learning rate in the algorithm (in the case where the randomness comes from the minibatches), at least for the oracles created from mini batches satisfy the assumptions required? Stated differently, if batch size is too small there seems to be no guarantee for the algorithm. What happens in the experiments if the batch size is made much smaller?*
>
> **Response 2**: Yes, indeed, our theory requires the minibatch size to be sufficiently large. However, in our experiments, even small fixed-size minibatches provide good oracles. In our experimental setting, changing the minibatch size from 128 to 32 (and keeping all other parameters the same) results in a very similar performance for both the kernel logistic regression instances and the MLP neural network model. For the convolutional neural network model, the performance of ALOE 2 is similar when the batch size is decreased to 32, while the performance of ALOE 1 degrades (it becomes comparable to the performance of SLS). This is potentially because ALOE 1 changes the step size more aggressively, and thus is more affected by a smaller batch size (which corresponds to less accurate estimates of the gradient / function value). Hence, in some cases one may want to adjust the gamma parameter accordingly if the batch size is decreased.
>
> ----
>
> **Question 3**: *The norms in equation 2, and in general, is that any norm? or maybe the two norm of the vector?*
>
> **Response 3**: Sorry, it is a 2-norm. We do not use any other norm in the paper. We will state this.
>
> ----
>
> **Question 4**: *Assumption 2, is that a standard assumption that the errors in different steps are independent (they could depend on the same x here)?*
>
> **Response 4**: The error distribution can depend on the same $x$, but the errors are indeed assumed to be independent on different steps, which is satisfied in the settings we consider. (For instance, for expected loss minimization in machine learning, this can be satisfied by choosing a different, independent sample set in each iteration.)
>
> ----
>
> **Question 5**: *Does the iteration bound (almost) match an obvious lower bound for this kind of oracles?*
>
> **Response 5**: We are not sure what lower bounds you mean exactly. There are lower bounds on the iteration complexity for deterministic oracles (see works of Nesterov, and Cartis, Gould and Point), which should also work as lower bounds for the iteration complexity using our more relaxed oracle. And indeed, our complexity bound almost matches those (up to a constant factor depending on the parameters of the probabilistic oracles).
>
> ----
>
> **Question 6**: *Can it be shown that for these kinds of oracles the recomputation of function value and gradient in each step in the line search is necessary in some sense (could the standard armijo SGD also converge here as that may be cheaper to run (data could be expensive to generate for instance with zeroth order optimization with finite difference approximation in reinforcement learning).*
>
> **Response 6**: In the case when the noise in the function value is bounded deterministically by $\epsilon_f$, it is not necessary to recompute the function value at the current point $f(x)$, but computation of $f(x^+)$ is still needed. However, $g_k$ has to be recomputed, because it may not be a descent direction; thus if not recomputed, the line search may either fail or produce a tiny step, which in turn will make the subsequent steps quite inefficient. In terms of theory, it is necessary to recompute $g_k$ because we need to ensure that true iterations happen often enough with sufficient probability. Furthermore, in deep neural network model training, the function values $f(x)$ (results of forward propagation) actually come for free as the gradients (backward propagation) are calculated, since any backward step needs to be preceded by a forward step.  Finally, as far as we know there is no algorithm as “standard Armijo SGD”.
>
> ----
>
> **Question 7**: *why use absolute values around |e(x)| on line 200 - 202, e(x) is already non-negative. Am I missing something there?*
>
> **Response 7**: Thank you, this is an error – an artifact of changing the definition for $e(x)$ at a late stage.

---

> > ### Comment · Reviewer_ApuJ · 2021-08-27
> > **update**
> >
> > I thank the authors for the response to my questions. I apologize if my questions were not completely clear. I was trying to ask if the upper bound shown is the best possible under the assumptions given, and as i understand your answer it basically is as  deterministic oracles may be a special case of the more general assumptions considered in the paper.  Similarly for question 6, i was trying to get an insight into how necessary the steps of the algorithm are, again given the assumptions used, and your answer makes good sense.

---

> > > ### Author Response · Authors · 2021-08-27
> > > **Thank you**
> > >
> > > Yes, thank you for reading the response.

---

### Official Review · Reviewer_V21B · 2021-07-15

**Rating:** 6
**Confidence:** 3

**Summary:**

This paper studied the line-search method with stochastic zeroth-order and first-order oracle. The author(s) extended prior works by (1) relaxing assumptions (2) providing a high probability bound (exponential tail bound) for the stopping time.

**Ethical Concerns:**

No ethical concern.


**Limitations And Societal Impact:**

This is a theoretical work and I do not see any limitation and negative social impact.


**Main Review:**

Pros:
- The structure of the paper is clear and most contents are easy to follow.
- Related works are well-addressed to my knowledge.

Cons:
- Eq (4) holds for the finite sum of convex functions and usually does not hold for nonconvex functions. Therefore I don’t think Proposition 2 could hold for nonconvex function. Assume that Proposition 2 is satisfied, then it means we have an accurate estimate of the true gradient at every saddle point (with prob $1 - \delta$). Can the author(s) provide a nonconvex example that satisfies Proposition 2? I understand that similar assumptions are also used in prior works, but I am confused about whether it is an appropriate assumption.
- One extension of this work to prior work is the $\epsilon_g$ used in eq. (2). But this also makes the theoretical result weaker, $\epsilon$ cannot be smaller than $\epsilon_g / \eta$ (see Assumption 4). I agree that this paper relaxed the assumptions (see eq. (1) and eq.(2)) used in prior works, but the obtained result is also weaker. I don’t think that the relaxed assumption is a significant contribution. The major contribution of this paper I think is the exponential high probability bound obtained (in contrast to the convergence in expectation in prior work). But I am not sure if this result is significant enough to be accepted at NeurIPS.

**Time Spent Reviewing:**

3 hours

---

> ### Author Response · Authors · 2021-08-10
> **Response to Reviewer V21B**
>
> Thank you for the review and questions!
>
> ----
>
> **Question 1**: *Eq (4) holds for the finite sum of convex functions and usually does not hold for nonconvex functions. Therefore I don’t think Proposition 2 could hold for nonconvex function. Assume that Proposition 2 is satisfied, then it means we have an accurate estimate of the true gradient at every saddle point (with prob $1-\delta$). Can the author(s) provide a nonconvex example that satisfies Proposition 2? I understand that similar assumptions are also used in prior works, but I am confused about whether it is an appropriate assumption.*
>
> **Response 1**: There is a key misunderstanding here. The right hand side of equation (4) contains two parts: $M_c$ and $M_v||\phi(x)||^2$. The second part vanishes when the gradient is zero (saddle point, for example), but not $M_c$. This is a very general bound on the variance which applies to most functions. It allows the variance to grow large (proportional to the gradient norm), but at the same time remain positive for any $x$. It is precisely the fact that we do **not** assume $M_c=0$ (unlike some other works, for example [VML+19]) that makes our analysis more general.
>
> ----
>
> **Question 2**: *One extension of this work to prior work is the $\epsilon_g$ used in eq. (2). But this also makes the theoretical result weaker, $\epsilon$ cannot be smaller than $\epsilon_g/\eta$ (see Assumption 4). I agree that this paper relaxed the assumptions (see eq. (1) and eq.(2)) used in prior works, but the obtained result is also weaker.*
>
> **Response 2**: The result is not weaker in any sense that we see. If $\epsilon_g$ is zero, the results are at least as strong as those in the prior literature (in fact they are simpler, with better constants than those, for example in [PS20]). If $\epsilon_g$ is not zero, this means that it is not possible in general to obtain an estimate $g$ of the gradient $\nabla \phi(x)$ with $||g - \nabla \phi(x)|| < \epsilon_g$. In particular, this allows the theory to capture settings where the gradient estimates are biased. This is, for instance, the case when (randomized) finite difference methods are applied to compute gradient estimates for noisy functions [BCCS21]. Naturally, if the gradient estimates are biased, convergence to an exact stationary point is impossible; one can only hope to converge to a neighborhood of a stationary point, where the size of the neighborhood depends on $\epsilon_g$ (and $\epsilon_f$). Our results show that the line search method and its convergence theory hold until that accuracy threshold is reached.

---

> > ### Comment · Reviewer_V21B · 2021-08-26
> > **Thanks for your clarification**
> >
> > I would like to thank the authors for their response. The explanation for Eq. (4) is very valid and I agree that this is a quite general condition that could also hold for nonconvex functions.
> >
> > For Theorem 1. when $\epsilon_g$ and $\epsilon_f$ are not close to 0, I think there is no guarantee for Algorithm 1 to converge to a solution with a small norm. But I think this is expected since we cannot expect linesearch-based method to converge when the noise of the gradient is too large.
> >
> > The author(s) addressed my major concerns and I am increasing my score.

---

> > > ### Author Response · Authors · 2021-08-27
> > > **Thank you**
> > >
> > > We appreciate you reading our response.

---

### Official Review · Reviewer_1krj · 2021-07-16

**Rating:** 6
**Confidence:** 3

**Summary:**

The paper proposed Adaptive Line-search with Oracle Estimations (ALOE) for nonconvex smooth stochastic optimization, the algorithm is based on modified Armijo condition. The algorithm allows the oracles to be biased and only requires the population function $\phi$ to be Lipschitz smooth and some other mild conditions on the stochastic process.

**Limitations And Societal Impact:**

This work does not involve issues related to the negative impact of society.

**Main Review:**

The paper aims to solve a practical problem in nonconvex optimization on parameter selection by modified Armijo line-search. The flow of the work is clear. Compared to previous closely related work [VML+19], it extend the setting to stochastic case with biased oracle, and provides a high probability complexity bound.

I have some confusions in this work, hope authors can add more discussion to address:
- In stochastic optimization, generally it will assume that the function/gradient estimation is unbiased with bounded variance, i.e. $\mathbb{E}f(x,\xi)=\phi(x)$, $\mathbb{E}g(x,\xi)=\nabla\phi(x)$, $\mathbb{E}||g(x,\xi)-\nabla \phi(x)||^2\leq\sigma^2$, instead of the norm form in Eq (1) and (2). So it seems that the main convergence result cannot recover the standard unbiased line-search algorithm result (if any, maybe $\mathcal{O}(\epsilon^{-4})$?), because $\epsilon_f=0$ will recover the deterministic line-search result $\mathcal{O}(\epsilon^{-2})$, is that correct, and is there any solution?
- Lower bound requirement on $\epsilon$. It seems that the setting in Eq (1) and (2) should contain the common unbiased setting above ($\mathbb{E}f(x,\xi)=\phi(x)$ but $\mathbb{E}|f(x,\xi)-\phi(x)|$ can still be upper bounded), but common stochastic optimization algorithm with unbiased oracle may not need the lower bound requirement in the accuracy $\epsilon$ (i.e. converge with arbitrary small $\epsilon$), is that correct? Even though chasing high accuracy may not be that important in machine learning, but as a theoretical paper in optimization, I am still wondering what is the case for high accuracy.
- In Assumption 2, it requires the function estimation error $e_i, e_i^+$ to be deterministically bounded by $\epsilon_f$ seems to be much stronger than boundedness in expectation in Eq (1) (authors also mentioned it is strong in Line 65), is that correct? Especially concerning that here the main theorem requires that the accuracy $\epsilon$ to be larger than $\text{poly}(\epsilon_f)$.

Based on the confusions above, currently I tend to reject, but after the rebuttal, I will appreciate the authors to address my confusion, and definitely reconsider my decision. Thank you.

**Time Spent Reviewing:**

6

---

> ### Author Response · Authors · 2021-08-10
> **Response to Reviewer 1krj**
>
> Thank you for the review and questions!
>
> ---
>
> **Question 1**: *In stochastic optimization, generally it will assume that the function/gradient estimation is unbiased with bounded variance, i.e. $\mathbb{E} f(x, \xi) = \phi(x)$, $\mathbb{E} g(x, \xi) = \nabla \phi(x)$, $\mathbb{E} ||g(x, \xi) - \nabla \phi(x)||^2 \leq \sigma^2$, instead of the norm form in Eq (1) and (2). So it seems that the main convergence result cannot recover the standard unbiased line-search algorithm result (if any, maybe $\mathcal{O}(\epsilon^{-4})$?), because $\epsilon_f = 0$
>  will recover the deterministic line-search result $\mathcal{O}(\epsilon^{-2})$, is that correct, and is there any solution?*
>
> **Response 1**: Thank you for this question! There is no variant of stochastic line search that is applied under the standard assumptions  (i.e. the gradient estimates are unbiased with bounded variance that does not go to zero when the gradient does), without reducing the variance.  In case when the unbiased gradient estimates (with bounded variance) are given, equation (2) can be ensured by choosing a sufficiently large minibatch and thus making the variance smaller. The total sample complexity is then going to be the sum of all minibatch sizes over all iterations. If we can show that the minibatch size is at most $\mathcal{O}(1/\epsilon^2)$, then the total sample complexity of our method will indeed be $\mathcal{O}(1/\epsilon^4)$ (just like that of SGD). Bounding the minibatch size from below is far from trivial, and it is in fact a subject of our ongoing work. But one can see where the result would come from -- the batch size needs to be inversely proportional to $(\alpha_k||g_k|| )^2$, and before the stopping time, $\alpha_k$ (as we show) tends to not be smaller than a constant ($\bar \alpha$), while $||g_k||^2$ tends to not be smaller than $\epsilon^2$. So in summary this algorithm is expected to have $\mathcal{O}(1/\epsilon^4)$ sample complexity under the standard unbiased gradient assumptions (ongoing work), while this current paper is an essential step toward showing this results but applies to more general settings.
>
> ----
>
> **Question 2**: *Lower bound requirement on $\epsilon$. It seems that the setting in Eq (1) and (2) should contain the common unbiased setting above ( $\mathbb{E} f(x, \xi) = \phi(x)$, but $\mathbb{E} |f(x, \xi) - \phi(x)|$ can still be upper bounded), but common stochastic optimization algorithm with unbiased oracle may not need the lower bound requirement in the accuracy $\epsilon$ (i.e. converge with arbitrary small $\epsilon$), is that correct? Even though chasing high accuracy may not be that important in machine learning, but as a theoretical paper in optimization, I am still wondering what is the case for high accuracy.*
>
> **Response 2**: Also a good question, thank you. So if $f(x,\xi)$ is an unbiased estimate of $\phi(x)$, then Eq (1) can have arbitrarily small $\epsilon_f$, by choosing a large enough minibatch size, and since the gradients are unbiased in the above setting, $\epsilon_g$ can be arbitrarily small as well. Thus, for any arbitrarily small $\epsilon$, the method can achieve this accuracy with an appropriate minibatch size.
>
> ----
>
> **Question 3**: In Assumption 2, it requires the function estimation error $e_i, e_i^+$ to be deterministically bounded by $\epsilon_f$ seems to be much stronger than boundedness in expectation in Eq (1) (authors also mentioned it is strong in Line 65), is that correct? Especially concerning that here the main theorem requires that the accuracy $\epsilon$  to be larger than $\text{poly}(\epsilon_f)$.
>
> **Response 3**: Assumption 2 states that **either** these estimates are deterministically bounded **or** they are independent random variables. If they are deterministically bounded, they are allowed to be even adversarial! This is still covered under Eq (1). On the other hand, if the errors are allowed to be arbitrarily large, then they need to be independent.

---

> > ### Comment · Reviewer_1krj · 2021-08-27
> > **Thank you for the response**
> >
> > Thank you for the detailed response, which basically solves my confusion, I am reconsidering my score. So the remedy should be a batch size.
> >
> > An additional problem: for response 2, you said with a large enough batch size, $\epsilon_f$ can be arbitrarily small, does it mean that, besides the bounded variance for the gradient estimation, you also need to add an additional assumption that the function value estimation $f(x,\xi)$ also has a bounded variance (i.e., $\mathbb{E}|f(x,\xi)-\phi(x)|^2\leq \sigma^2$)?
> >
> > Also, just for clarification, you mentioned the error can be allowed to be adversarial, I know "adversarial" in adversarial machine learning, which, for example, adds a malignant noise to the picture (i.e., the variable $x$) to fail the classifier. Is there any example for your adversarial error? (the different seems to lie in that the error is added to $x$ (the picture) and $\phi(x)$ (the objective) here)
> >
> > Thank you.

---

> > > ### Author Response · Authors · 2021-08-27
> > > **Clarification response**
> > >
> > > Yes, indeed we assume the variance of $f(x,\xi)$ is bounded, but this assumption is already implied by the assumption that the error between
> > > $f(x,\xi)$ and $\phi(x)$ is subexponential.
> > >
> > > As for the adversarial error, yes, indeed we mean the error in the function $\phi$. So this error may be not random, for example, when $\phi(x)$ is computed via a simulation code based on some space discretization, then the computed values $f(x,\xi)$ will give an approximation of $\phi(x)$, but possibly not coming from a probability distribution. So, when we say "adversarial" we really mean that  all we know is that it can be as large as $\epsilon_f$  for any $x$. We can think of the following (somewhat artificial) scenario that actually involves an adversary - assume that $\phi(x)$ is the expected error of a binary classifier and  each time we try to compute this error (based on some sample set) the adversary flips a small percentage $\alpha$ of the labels. Then our error estimate is wrong by at most $\alpha$.

---

> > > > ### Comment · Reviewer_1krj · 2021-08-27
> > > > **Thank you**
> > > >
> > > > Thank you for the response. I decide to increase my score.
> > > >
> > > > Also, as the authors mentioned in previous response, I look forward to the extended version of the current submission with more discussion on batch size.

---

### Official Review · Reviewer_EQ5H · 2021-07-18

**Rating:** 7
**Confidence:** 4

**Summary:**

Consider unconstrained smooth optimization with probabilistic zeroth- and first-order oracles. This paper proposes a new gradient descent with line search method and provides a high-probability upper bound of the iteration complexity. The authors highlight that their assumptions on the noisy zeroth- and first-order oracles are weaker than those in literature.

**Limitations And Societal Impact:**

Please address my comments in the main review.

**Main Review:**

I like the idea of querying new noisy gradients even when the Armijo condition is not met in the proposed line search algorithm. I feel this is perhaps the natural approach in the stochastic setting.

Below are some comments.
- The numerical experiment does not exactly follow the theory: In the numerical experiment, the mini-batch sizes are fixed, first- and zero-th order oracles use the same minibatch and $\gamma_{\text{dec}} \neq 1 / \gamma_{\text{inc}}$. Why is the theory not followed? What is the empirical result when the theory is strictly followed?
- It is assumed that the the error of the probabilistic first-order oracle is bounded from above by the norm of the noisy gradient (scaled by a constant) with high probability. This assumption seems to suggest that the mini-batch size cannot be a constant but should increase when the iterates approach a stationary point. Is this increasing mini-batch size necessary?
- The authors highlight that unlike in existing literature, they do not assume the noisy gradient to be Lipschitz. When is this relaxation of the assumption significant?
- Line 63--88 addresses related existing results, arguing existing results are restrictive and do not provide high-probability guarantees. It would be good if the authors can clarify the key idea that enables them to make the breakthrough in this paper.
- Line 293: Why is resetting the step size in each iteration is "impractical"?

The presentation is clear. Below are some comments on the presentation:
- Line 85: Saying the existing assumptions are "very restrictive" is vague. In what sense are they restrictive?
- Line 152--156: The definitions of $I_k$, $\Theta_k$, and $Z_k$ are not used in Section 3. I think their appearance should be postponed to Section 4.
- Line 240: The name of the proposed algorithm should be appear in the second last section.

**Time Spent Reviewing:**

5

---

> ### Author Response · Authors · 2021-08-10
> **Response to Reviewer EQ5H**
>
> Thank you for the review and questions!
>
> ---
>
> **Question 1**: *The numerical experiment does not exactly follow the theory: In the numerical experiment, the mini-batch sizes are fixed, first- and zero-th order oracles use the same minibatch and $\gamma_{dec} \neq 1/\gamma_{inc}$. Why is the theory not followed? What is the empirical result when the theory is strictly followed?*
>
> **Response 1**:
> If we perform the experiments exactly as the theory says with $\gamma_{dec}=0.8$ and $\gamma_{inc}=1.25$, the results are actually very similar to the plots we had in the paper. ALOE still achieves better training loss than the SLS algorithm in 62 out of 64 datasets, and is always better than the full gradient line search. However, we decided to do the experiments with $\gamma_{dec}=0.7$ and $\gamma_{inc}=1.25$ because the results with those parameters were slightly better. Essentially, our experiments suggest that choosing $\gamma_{dec}$ to be equal to the inverse of $\gamma_{inc}$ will give good results, but there may be gains from having another degree of freedom in terms of choosing $\gamma_{dec}$ to be different from the inverse of $\gamma_{inc}$ in practice.
>
> In fact the theory can be applied to the case when $\gamma_{dec}$ and $\gamma_{inc}$ are not equal (see [GRVZ18]), but the theory and the notation will need to be a lot messier and harder to follow and some small modifications to the algorithm will have to be imposed. The key property is that when $\alpha_k$ is small enough it tends to go up, and this can be accomplished with different values of $
> \gamma_{dec}$ and $\gamma_{inc}$ but then one has to be careful with bounding the number of large and small steps.
>
> Theory indeed dictates possibly not fixed minibatch sizes, with first- and zeroth-order oracles using independent minibatches. However, these are required to guarantee that the oracles behave they way we need them to. What our experiments show is that, in practice, oracles based on fixed minibatch sizes (and using the same minibatch), often behave as well as our theory requires.
>
> ----
>
> **Question 2**: *It is assumed that the the error of the probabilistic first-order oracle is bounded from above by the norm of the noisy gradient (scaled by a constant) with high probability. This assumption seems to suggest that the mini-batch size cannot be a constant but should increase when the iterates approach a stationary point. Is this increasing mini-batch size necessary?*
>
> **Response 2**: In some cases it is necessary and in some it is not. Generally, it may be necessary to reduce the variance of the gradient estimates as the algorithm progresses to achieve the presented complexity rates. In general, the variance can be reduced by increasing the minibatch size, but in the case when the “interpolation” condition holds, the variance even with a fixed batch size, is reduced proportional to the size of the gradient. This is precisely why we use a general definition of the oracle and then discuss how it can be implemented in Section 2. In Section 2, we derive explicit conditions for different cases on the sample size. For example, if $M_c$ (the bound on the variance that does not depend on the gradient norm) is not 0 (no interpolation) then if we limit the size of the minibatch, it results in convergence only up to an $\epsilon$ bounded from below by a quantity dependent on the variance and the minibatch size.
>
>  ----
>
> **Question 3**: *The authors highlight that unlike in existing literature, they do not assume the noisy gradient to be Lipschitz. When is this relaxation of the assumption significant?*
>
> **Response 3**: This is actually significant in many ML settings. For example, in the paper “On the Local Minima of the Empirical Risk“ by Jin, Ge, Liu and Jordan, it is shown that for the ReLU function, the population risk ($\phi(x)$) is generally better behaved than empirical risk ($f(x, \xi)$) and specifically it is smooth when the data distribution is Gaussian, while clearly no empirical risk based on ReLU is Lipschitz smooth. Moreover, even if the empirical loss is  Lipschitz smooth, the constant might vary widely by the minibatch. For example, in the case of the logistic loss, the Lipschitz constant of the loss measured on a misclassified data point is much larger than that of a correctly classified one.
>
> ----
>
> **Question 4**: *Line 63--88 addresses related existing results, arguing existing results are restrictive and do not provide high-probability guarantees. It would be good if the authors can clarify the key idea that enables them to make the breakthrough in this paper.*
>
> **Response 4**: Thank you for this interested question. The analysis and the algorithm in [VML+19] relies on standard deterministic line search arguments, so it is very different in terms of analysis.  As for the other references, the key idea that enables the analysis here is the way that the iterations are counted. The use of Lemma 1 enables the high probability result, and is applied by utilizing an indicator random variable $U_k$ (large/small steps) which allow us to bound different types of iterations with respect to each other, using combinatorial arguments. Another key idea is the more general but properly defined zeroth order oracle and the subsequent definition of a true iteration ($I_k=1$).
>
> ----
>
> **Question 5**: *Line 293: Why is resetting the step size in each iteration is "impractical"?*
>
> **Response 5**: This comment applies to standard line search, where multiple backtracking steps are performed along the same direction. If the step size is reset to a fixed value at each iteration then to what value should it be reset to? This value is a parameter which may be chosen too large – which means that each step will result in many backtracking steps, or too small – which means that the line search will never be able to take large steps even though they could lead to rapid progress. This is fairly well known in deterministic optimization – the effective step size parameter value tends not to vary too much between neighboring iterations, and thus it is much more practical to allow it to adjust only by a constant multiple between steps.
>
> ----
>
> **Question 6**: *Line 85: Saying the existing assumptions are "very restrictive" is vague. In what sense are they restrictive?*
>
> **Response 6**: Firstly, there is an assumption that the variance of the stochastic gradient reduces to zero at optimality – this is very restrictive and does not hold in many situations. Secondly the analysis is only applied to a finite sum of Lipschitz smooth functions. Lastly (and perhaps most importantly), in the nonconvex case there is an assumption that the step size parameter is bounded above by $1/LM_v$ (in the notation of our paper), which essentially removes the advantage of line search, since the step size is chosen at least as conservatively as it would be without line search. We will try to find space to clarify this.
>
> ----
>
> **Question 7**: *Line 152--156: The definitions of $I_k$, $\Theta_k$, and $Z_k$ are not used in Section 3. I think their appearance should be postponed to Section 4.*
>
> **Response 7**:
> This is true, we felt it was better to define these notions right after the definition of the filtration, but we are happy to reconsider.
>
> ----
>
> **Question 8**: *Line 240: The name of the proposed algorithm should be appear in the second last section.*
>
> **Response 8**: Thank you, yes.

---

> > ### Comment · Reviewer_EQ5H · 2021-08-29
> > **Increase my evaluation to 7**
> >
> > The answers to my questions are satisfactory to me (except the answer for the fourth one, which I think is too detailed and lacks a high-level insight). I think the simple and elegant way of implementing line search in stochastic methods is inspiring, though it seems not what the authors want to emphasize in the title. I increase my rating to 7.

---

> > > ### Author Response · Authors · 2021-08-30
> > > **Thank you**
> > >
> > > We appreciate you reading our response.

---

### Decision · Program_Chairs · 2021-09-27

**Decision:**

Accept (Poster)

**Comment:**

This was perhaps the most difficult decision in my pile, and I felt the need to carefully reviewed the paper myself. The main point against the paper is that the results are similar to many existing results that hold in expectation, and it is unlikely that members of the community would be surprised that this algorithm works with high probability. On the other hand, the authors have done a thorough job both theoretically and empirically. I believe this work will make a good reference on the topic, and the authors have addressed nearly all the points brought up in the reviews (e.g., experiments on non-convex problems and issues with the assumptions). So overall I leaned towards recommending acceptance.

Some additional "reviewer comments" from my readthrough:

- The paper needs to spend more time, especially early in the paper, making connections to machine learning. For example, the paper needs to be more clear that in order to converge the mini-batch size must grow as a stationary point is approached. It should discuss the relationship between the mini-batch size and the error in the gradient (this was brought up in several reviews, and this discussion will be useful for many readers).

- I feel like this algorithm is more-closely related to "growing batch" methods than to stochastic gradient methods. The paper should discuss these related works, and emphasize the connections to them. The paper could also refer to the growing-batch papers as ways people have tried to set the batch size in methods like this (because setting the batch size for hybrid methods like this is not easy, so referring to other papers is probably all you will have space for).

- The paper needs to *explicitly say that there is a minimum batch size required*, given an accuracy level, which is different than the usual SGD setting.

- Given the connection, I would also like to see a growing-batch method in the experiments (SLS does not quite seem like the right method to compare to).